# VT-DUDA: Visual Token Conditioning for Diffusion-guided Unsupervised Domain Adaptation

**Xuan Qi**[*]                                                                 *xuan.qi@iit.it*
*AI for Good (AIGO), Istituto Italiano di Tecnologia*
*DITEN, University of Genoa*

**Daniele Berardini**                                                          *daniele.berardini@iit.it*
*AI for Good (AIGO), Istituto Italiano di Tecnologia*

**Dario Serez**                                                                *dario.serez@iit.it*
*AI for Good (AIGO), Istituto Italiano di Tecnologia*

**Vito Paolo Pastore**                                                         *vito.paolo.pastore@unige.it*
*AI for Good (AIGO), Istituto Italiano di Tecnologia*
*MaLGa, DIBRIS, University of Genoa*

**Vittorio Murino**                                                            *vittorio.murino@iit.it*
*AI for Good (AIGO), Istituto Italiano di Tecnologia*
*Department of Computer Science, University of Verona*

**Reviewed on OpenReview:** *https://openreview.net/forum?id=Y956680PCe*

## Abstract

Unsupervised domain adaptation (UDA) aims to learn a target-domain classifier from labeled source data and unlabeled target data under distribution shift. Recent diffusion-based UDA methods approach this problem by synthesizing labeled target-style images and training on the resulting synthetic data. However, their performance depends heavily on the conditioning design: class prompts provide only coarse guidance, while domain adaptation modules mainly control appearance, which may leave target-style synthesis insufficiently specified. We propose VT-DUDA, a visual-token conditioning framework for diffusion-guided UDA. Instead of relying only on text prompts, VT-DUDA uses source images to provide additional instance-level visual context for target-style synthesis. Specifically, VT-DUDA maps each source image to a compact sequence of visual tokens and forms a hybrid conditioning context by concatenating these tokens with the corresponding text embeddings along the cross-attention context dimension of a latent diffusion model. This provides instance-dependent conditioning beyond text alone, while synthesis is performed with the target-domain adapter branch. Because guidance is represented explicitly as a token sequence, the same interface also permits inference-time manipulation of the conditioning signal through token selection and token-strength adjustment. The proposed method preserves the standard diffusion objective and can be integrated into existing adapter-based diffusion frameworks without modifying the backbone. Across Office-31, Office-Home, and VisDA-2017, VT-DUDA improves average target-domain accuracy over strong discriminative and diffusion-based UDA baselines. The results suggest that, in generation-based UDA, a stronger conditioning interface can improve the downstream usefulness of synthetic target-style data. The project page is available at `https://xuanqi99.github.io/VT-DUDA/`.

---

[*]Corresponding author.

# 1 Introduction

Unsupervised domain adaptation (UDA) aims to learn a target-domain classifier from labeled source data and unlabeled target data under distribution shift. Classical UDA methods address this problem primarily in feature space through discrepancy minimization Li et al. (2020); Long et al. (2015; 2018), adversarial alignment Ganin et al. (2016); Saito et al. (2018), or prediction- and entropy-based regularization Xu et al. (2019); Rangwani et al. (2022); Zhang et al. (2023b; 2019); Xia et al. (2022); Wu et al. (2025); Jin et al. (2020). More recently, diffusion-based approaches Zhuang et al. (2024); Zhang et al. (2025); Rombach et al. (2022); Ho et al. (2020); Nichol & Dhariwal (2021); Song et al. (2020b); Song & Ermon (2020); Dhariwal & Nichol (2021) have explored a data-centric alternative: instead of aligning source and target features directly, they adapt a generative model to synthesize labeled target-style images and then train a classifier on the resulting synthetic set. In this view, the quality of the constructed synthetic supervision becomes an important factor in downstream adaptation.

Latent diffusion models are a natural choice in this setting because they support flexible conditioning through cross-attention Nichol et al. (2022); Podell et al. (2024); Rombach et al. (2022); Vaswani et al. (2017); Ho & Salimans (2022); Zhang et al. (2023a); Ye et al. (2023). In principle, target-style synthesis can be guided by class prompts, domain indicators, or lightweight adaptation modules. In practice, however, diffusion-guided UDA often relies on conditioning that maps distinct source instances of the same class to the same class-level text signal. That is, on the source side, multiple labeled source images from the same class are typically associated with the same class-level text condition (e.g., "a bed"), even though different source exemplars may contain different visual details that could be useful for target-style synthesis. On the target side, when target labels are unavailable, the text condition may be reduced to a generic prompt such as "an image." As a result, the denoiser may be trained under conditions that provide limited instance-level guidance for synthetic data construction.

In this paper, we study the role of conditioning specificity in diffusion-guided UDA and propose **VT-DUDA**, a **V**isual-**T**oken conditioning framework for **D**iffusion-guided **U**nsupervised **D**omain **A**daptation. The core idea is to augment text conditioning with instance-level visual tokens extracted from individual images. Specifically, given an image, VT-DUDA maps it to a compact sequence of visual tokens and concatenates these tokens with text embeddings through the standard cross-attention interface of a latent diffusion model. On the source side, the visual tokens are paired with class prompts, so that different labeled source instances sharing the same class label are not represented by exactly the same conditioning context. In this way, the class prompt specifies the category, while the visual tokens provide instance-specific information associated with the current source sample. On the target side, where class labels are unavailable, the visual tokens are paired with a generic prompt such as "An image," providing additional instance-level context beyond text alone for unlabeled target denoising.

A key design choice is to use a token-augmented cross-attention format on both domains. During training, the source branch receives class prompts paired with source-image visual tokens, whereas the target branch receives a generic target prompt paired with target-image visual tokens. During generation, we keep the same token-augmented cross-attention format but populate it with the source class prompt and the source-image visual tokens, and synthesize under the target branch. Thus, VT-DUDA aligns the conditioning interface between training and generation, while the semantic content of the context intentionally changes from target-image tokens during target denoising to source-image tokens during target-style synthesis. This design exposes the target branch to token-conditioned target-domain denoising and then uses source-conditioned tokens to provide instance-level guidance for labeled target-style generation.

This perspective motivates evaluating diffusion-guided UDA by the downstream usefulness of the generated dataset. The goal is not only to generate plausible target-style images, but to construct synthetic data that is useful for downstream target-domain classification. We therefore focus on the *utility* of the generated dataset for adaptation. In particular, when the conditioning interface is weak, increasing the generation budget alone may not necessarily improve downstream performance. VT-DUDA is designed to increase conditioning specificity within each class, with the goal of improving the downstream usefulness of synthetic target-style data under limited generation budgets.

The tokenized design also has two practical properties. First, because visual guidance is represented explicitly as a sequence of tokens, it permits inference-time manipulation through token subset selection and token-strength scaling. Second, the proposed conditioning interface remains compatible with inversion-based translation procedures such as DDIM inversion Song et al. (2020a); Mokady et al. (2023); Meng et al. (2021), allowing token-conditioned synthesis and translation-based augmentation to be used within the same framework.

Our contributions are summarized as follows:

- We propose **VT-DUDA**, a visual-token conditioning framework for diffusion-guided unsupervised domain adaptation that addresses insufficient instance-level guidance in existing diffusion-based UDA methods. Our method augments the standard cross-attention interface of an adapter-based latent diffusion model with instance-dependent visual tokens, enriching source-side class prompts and target-side generic prompts without modifying the diffusion backbone, VAE, or denoising objective.

- Within this framework, we study two token-conditioned routes for constructing synthetic target-style training data for UDA: pure-noise target-style synthesis and DDIM-inversion-based target-style translation. Both routes use the same visual-token conditioning interface, providing a unified way to generate labeled target-style samples from source-domain supervision under different generation protocols.

- Experiments on Office-31, Office-Home, and VisDA-2017 show that VT-DUDA improves average downstream target-domain accuracy over strong discriminative and diffusion-based baselines. We also observe that the tokenized conditioning interface supports inference-time token manipulation, and that combining pure-noise synthesis with inversion-based translation can provide additional gains in our setting.

## 2 Related Work

**Unsupervised domain adaptation.** Unsupervised domain adaptation (UDA) seeks to transfer knowledge from a labeled source domain to an unlabeled target domain under distribution shift. A large body of prior work addresses this problem in feature space through discrepancy minimization Li et al. (2020); Long et al. (2015; 2018), adversarial alignment Ganin et al. (2016); Saito et al. (2018), or prediction- and entropy-based regularization Xu et al. (2019); Rangwani et al. (2022); Zhang et al. (2023b; 2019); Xia et al. (2022); Wu et al. (2025); Jin et al. (2020). These approaches typically aim to learn domain-invariant representations for a fixed discriminative model. In contrast, generative approaches revisit UDA from a data-centric perspective, where adaptation is achieved by constructing target-style training data and then learning a classifier from the resulting synthetic domain.

**Diffusion-based adaptation under domain shift.** Diffusion models have recently been explored as tools for UDA and related cross-domain learning problems Zhuang et al. (2024); Zhang et al. (2025); Rombach et al. (2022). Rather than aligning source and target features directly, these methods leverage the generative capacity of latent diffusion models to synthesize target-style images or translate source images toward the target domain, after which downstream supervised learning is performed on the generated data. This line of work benefits from two properties of modern diffusion systems: pretrained generative priors for modeling appearance changes across domains Ho et al. (2020); Nichol & Dhariwal (2021); Song et al. (2020b); Dhariwal & Nichol (2021), and parameter-efficient adaptation through mechanisms such as adapters and LoRA Hu et al. (2022); Houlsby et al. (2019); Karimi Mahabadi et al. (2021); Chen et al. (2022). Our work belongs to this family of diffusion-guided UDA methods, but focuses on a different design axis: the conditioning interface used to construct synthetic supervision.

**Conditioning in diffusion models.** An important property of latent diffusion models is their flexible conditioning interface. Through cross-attention, the denoiser can incorporate signals such as text prompts, learned embeddings, or auxiliary context tokens Nichol et al. (2022); Podell et al. (2024); Rombach et al. (2022); Vaswani et al. (2017); Zhang et al. (2023a); Ye et al. (2023); Hertz et al. (2022). Existing conditioning

mechanisms are primarily developed for general-purpose generation, editing, or style control, where the main goals are controllability, personalization, or fidelity Garibi et al. (2025); Gal et al. (2023); Ruiz et al. (2023); Shah et al. (2024); Kumari et al. (2023); Zhang et al. (2023a); Ye et al. (2023); Hertz et al. (2022); Mokady et al. (2023); Brooks et al. (2023); Wei et al. (2023). In diffusion-guided UDA, however, the objective is different: the generated images should serve as *useful labeled training data* for downstream adaptation. Conditioning quality is therefore task-dependent. A conditioning mechanism that is useful for generic generation may not be the one that yields the most useful synthetic target-style samples for downstream adaptation.

Motivated by this distinction, we study how visual tokens can enrich the conditioning signal in diffusion-guided UDA. VT-DUDA represents each image as a compact sequence of visual tokens and defines the cross-attention context as the concatenation of the text-token sequence and the visual-token sequence. Fine-tuning and generation use the same token-augmented cross-attention format, although the semantic content of the conditioning context differs across stages. During fine-tuning, source samples are denoised with class prompts and source-image tokens, while target samples are denoised with a generic prompt and target-image tokens. During generation, source-image tokens are paired with the source class prompt and passed to the target branch to provide instance-level guidance for target-style synthesis. The resulting design is easy to integrate into adapter-based diffusion adaptation frameworks and allows inference-time manipulation through token selection and token-strength scaling.

## 3 Methodology

### 3.1 Problem Setup

We consider the standard UDA setting with a labeled source domain and an unlabeled target domain. Let $\mathcal{D}_{\mathrm{s}} = \{(x_i^{\mathrm{s}}, y_i^{\mathrm{s}})\}_{i=1}^{N_{\mathrm{s}}}$ denote the source dataset, where $x_i^{\mathrm{s}} \in \mathbb{R}^{H \times W \times 3}$ is an RGB image and $y_i^{\mathrm{s}} \in \{1, \ldots, C\}$ is its class label over $C$ shared categories. Let $\mathcal{D}_{\mathrm{t}} = \{x_j^{\mathrm{t}}\}_{j=1}^{N_{\mathrm{t}}}$ denote the unlabeled target dataset, drawn from a different image distribution but sharing the same label space. Equivalently, the two domains can be viewed as samples from distributions $\mathbb{P}_{\mathrm{s}}(X, Y)$ and $\mathbb{P}_{\mathrm{t}}(X, Y)$ defined on a common label space, where target labels are unavailable during training. Our goal is to learn a classifier that performs well on the target domain.

We adopt a generation-based view of UDA. Rather than aligning source and target features directly, we first adapt a conditional diffusion model using labeled source images and unlabeled target images, then synthesize a labeled target-style training set, and finally train a discriminative classifier on the resulting data. In this pipeline, the diffusion model is used to generate target-style samples conditioned on labeled source inputs, while the downstream UDA objective is applied at the classifier-training stage.

### 3.2 Preliminaries: Adapter-based Latent Diffusion

**LoRA adaptation.** Following LoRA Hu et al. (2022), given a frozen weight matrix $W_0 \in \mathbb{R}^{m \times n}$, LoRA parameterizes a low-rank update

$$\Delta W = W_{\mathrm{up}} W_{\mathrm{down}}, \tag{1}$$

where $W_{\mathrm{up}} \in \mathbb{R}^{m \times r}$, $W_{\mathrm{down}} \in \mathbb{R}^{r \times n}$, and $r \ll \min(m, n)$. For an input feature $x \in \mathbb{R}^n$, the adapted forward pass is

$$y = W_0 x + \lambda \Delta W x = W_0 x + \lambda W_{\mathrm{up}} W_{\mathrm{down}} x, \tag{2}$$

where $\lambda > 0$ is a scaling factor.

**Latent diffusion objective.** Let $E_{\mathrm{VAE}}$ and $D_{\mathrm{VAE}}$ denote the encoder and decoder interfaces of a pre-trained VAE Kingma & Welling (2013), both frozen throughout training. For notational simplicity, we absorb the latent scaling factor used by the pretrained latent diffusion model into these interfaces. A training image $x_0$ is mapped to a latent code $z_0 = E_{\mathrm{VAE}}(x_0) \in \mathbb{R}^{h \times w \times d}$. Given a diffusion timestep $\tau \in \{1, \ldots, T\}$, the forward noising process is

$$z_\tau = \sqrt{\bar{\alpha}_\tau} \, z_0 + \sqrt{1 - \bar{\alpha}_\tau} \, \epsilon, \qquad \epsilon \sim \mathcal{N}(0, I), \tag{3}$$

where $\bar{\alpha}_\tau = \prod_{k=1}^{\tau}(1 - \beta_k)$ under a variance schedule $\{\beta_k\}_{k=1}^T$. We also set $\bar{\alpha}_0 = 1$, so that $z_0$ corresponds to the clean latent endpoint used in sampling and inversion schedules.

Let $\epsilon_\theta(\cdot)$ denote the U-Net noise predictor with parameters $\theta$. Given a text prompt $p$, a frozen text encoder produces a token sequence $e(p)$, which is passed to the U-Net through cross-attention. Standard adapter-based fine-tuning minimizes the denoising objective Ho et al. (2020); Nichol & Dhariwal (2021); Song et al. (2020b); Song & Ermon (2020)

$$\mathcal{L}_{\text{diff}}(\Delta\theta) = \mathbb{E}_{z_0,\tau,\epsilon,p}\Big[\|\epsilon - \epsilon_{\theta_0+\Delta\theta}(z_\tau, \tau, e(p))\|_2^2\Big], \tag{4}$$

where $\theta_0$ denotes frozen backbone parameters and $\Delta\theta$ denotes trainable adapter parameters.

**Domain-specific adapter branches.** To model source- and target-domain appearance within a shared diffusion backbone, we use two domain-specific adapter branches indexed by a binary variable $\sigma \in \{0, 1\}$, where $\sigma = 0$ denotes the source branch and $\sigma = 1$ denotes the target branch. For each adaptable weight matrix, the active adapted weight is written as

$$W^{(\sigma)} = W_0 + \lambda\Delta W^{(\sigma)}. \tag{5}$$

Here, $\Delta W^{(\sigma)}$ denotes the effective low-rank update selected by branch $\sigma$. This notation describes the branch-dependent update at the functional level; the two effective updates do not have to be parameterized by two fully independent LoRA modules. In our implementation, described in Appendix A.1, the source and target effective updates are realized through a shared low-rank basis and a target-activated modulation matrix. We use $\Delta\theta$ to denote the collection of all trainable adapter parameters. The same domain indicator $\sigma$ is used consistently during diffusion training and sample generation.

### 3.3 Visual Token Conditioning

We introduce a visual-token conditioning interface that augments text prompts and domain control with additional instance-specific visual features.

At a high level, VT-DUDA consists of three stages. First, we jointly train a domain-adaptive diffusion model and an image-to-token encoder using labeled source images and unlabeled target images. Second, for each labeled source instance, we extract visual tokens from the source image and combine them with the class prompt to synthesize labeled target-style samples under the target branch of the diffusion model. Third, we train a downstream classifier on the resulting synthetic target-style dataset, optionally together with an unsupervised target regularizer. The same tokenized conditioning interface can also be used for inference-time token manipulation and for inversion-based translation.

#### 3.3.1 Image-to-Token Encoder

We introduce an image-conditioned encoder $G_\psi$ that maps an input image $v \in \mathbb{R}^{H \times W \times 3}$ to a sequence of $M$ visual tokens,

$$U = G_\psi(v) = [u_1; \ldots; u_M] \in \mathbb{R}^{M \times d_c}, \qquad u_m \in \mathbb{R}^{d_c}, \tag{6}$$

where $d_c$ denotes the cross-attention context dimension of the diffusion model.

$G_\psi$ first extracts a global visual representation from the input image using a ResNet-based backbone. This representation is then mapped to an $M$-dimensional latent code $a(v) \in \mathbb{R}^M$, and each scalar component $a_m(v)$ is transformed into a $d_c$-dimensional visual token through a token-specific mapping $g_m$:

$$u_m = g_m\big(a_m(v)\big), \qquad m = 1, \ldots, M. \tag{7}$$

The resulting token sequence $U$ is concatenated with text embeddings and used as additional conditioning in cross-attention.

This parameterization converts image-level information into a fixed number of conditioning tokens. The resulting token sequence $U$ is concatenated with text embeddings and used as additional conditioning in

cross-attention. The encoder is trained jointly with the diffusion model using only the denoising objective, without auxiliary token-level supervision.

$$U_i^s = G_\psi(x_i^s), \qquad U_j^t = G_\psi(x_j^t). \tag{8}$$

### 3.3.2 Hybrid Cross-Attention Context

Let $e(p) \in \mathbb{R}^{M_p \times d_c}$ denote the token-wise embedding sequence of a text prompt $p$. For a labeled source sample $(x_i^s, y_i^s)$, we use a class prompt $p_s(y_i^s)$, e.g., "A [class]". For a target sample $x_j^t$, we use a generic target prompt $p_t$, e.g., "An image".

We write

$$R_i^s = e\big(p_s(y_i^s)\big) \in \mathbb{R}^{M_p^s \times d_c}, \qquad R_j^t = e(p_t) \in \mathbb{R}^{M_p^t \times d_c}. \tag{9}$$

The source and target conditioning contexts are then formed by concatenating text tokens and visual tokens along the sequence dimension:

$$\begin{aligned}
S_i^s &= \big[R_i^s \,;\, \kappa_s U_i^s\big] \in \mathbb{R}^{(M_p^s + M) \times d_c}, \\
S_j^t &= \big[R_j^t \,;\, \kappa_t U_j^t\big] \in \mathbb{R}^{(M_p^t + M) \times d_c},
\end{aligned} \tag{10}$$

where $\kappa_s, \kappa_t > 0$ control the relative strength of visual-token guidance in the source and target branches, respectively.

This design leaves the denoiser architecture and diffusion objective unchanged, and modifies only the conditioning sequence passed to cross-attention. On the source side, the visual tokens make the conditioning depend on the current labeled sample in addition to its class prompt. On the target side, the generic prompt provides limited class information; adding target-image tokens trains the target branch under a token-augmented cross-attention format. At generation time, the same format is reused, but the context is populated with a source class prompt and source-image visual tokens. Thus, the training and generation stages share the same conditioning interface, while the image source and prompt specificity of the conditioning content differ by design.

### 3.3.3 Joint Diffusion Training

We jointly optimize the domain-specific adapter parameters $\Delta\theta$ and the token encoder parameters $\psi$. For each labeled source sample $(x_i^s, y_i^s) \sim \mathcal{D}_s$, we construct the source conditioning context $S_i^s$ using the class prompt $p_s(y_i^s)$. For each unlabeled target image $x_j^t \sim \mathcal{D}_t$, we construct the target conditioning context $S_j^t$ using the generic prompt $p_t$.

Let

$$z_{i,0}^s = E_{\text{VAE}}(x_i^s), \qquad z_{j,0}^t = E_{\text{VAE}}(x_j^t) \tag{11}$$

denote the corresponding latent codes, and let $z_{i,\tau}^s$ and $z_{j,\tau}^t$ be their noised versions obtained through Equation 3. We denote by

$$\epsilon_{\theta_0 + \Delta\theta}(z, \tau, S, \sigma) \tag{12}$$

the conditioned noise predictor, where $S$ is the hybrid conditioning context and $\sigma \in \{0, 1\}$ selects the active domain-specific adapter branch.

The joint diffusion training objective is

$$\begin{aligned}
\mathcal{L}_{\text{train}}(\Delta\theta, \psi) = {} & \lambda_s \, \mathbb{E}_{(x_i^s, y_i^s) \sim \mathcal{D}_s, \, \tau, \, \epsilon}\Big[\big\|\epsilon - \epsilon_{\theta_0 + \Delta\theta}(z_{i,\tau}^s, \tau, S_i^s, 0)\big\|_2^2\Big] \\
& + \lambda_t \, \mathbb{E}_{x_j^t \sim \mathcal{D}_t, \, \tau, \, \epsilon}\Big[\big\|\epsilon - \epsilon_{\theta_0 + \Delta\theta}(z_{j,\tau}^t, \tau, S_j^t, 1)\big\|_2^2\Big],
\end{aligned} \tag{13}$$

where $\lambda_s, \lambda_t > 0$ balance the source and target denoising terms. In our default setting, source and target mini-batches are sampled with equal frequency and we use $\lambda_s = \lambda_t = 1$. Unless otherwise stated, $\tau$ is sampled uniformly from $\{1, \dots, T\}$ and $\epsilon \sim \mathcal{N}(0, I)$.

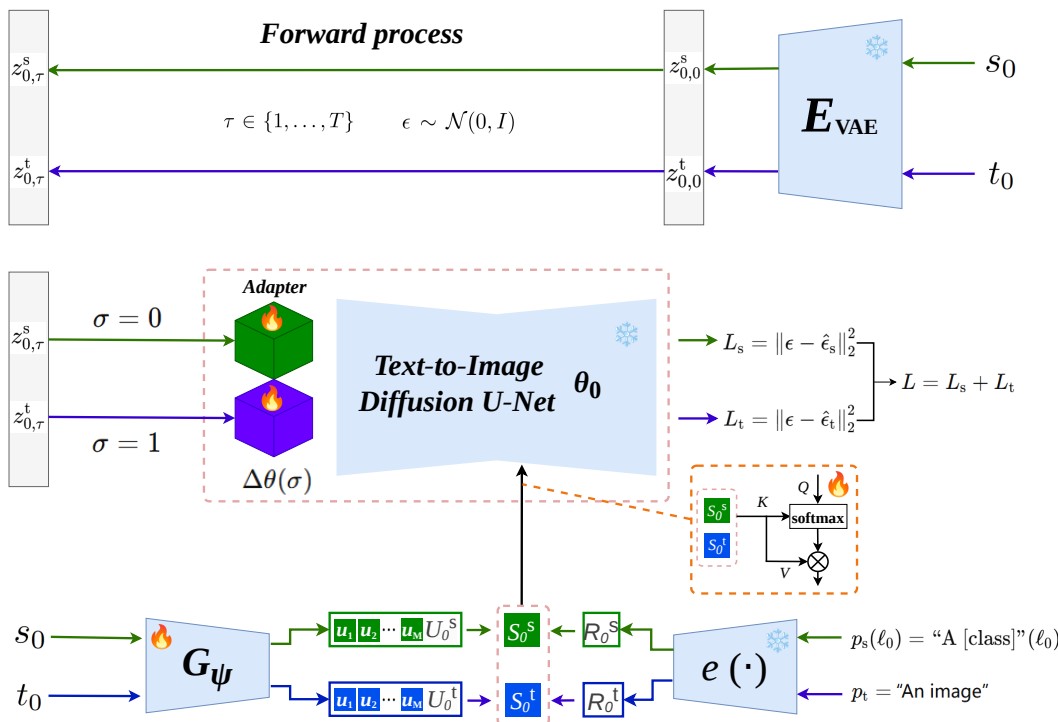

Figure 1: Overview of VT-DUDA. We jointly train domain-specific diffusion adapters and an image-to-token encoder on source and target images.

The first term optimizes denoising on labeled source samples under class-conditioned, exemplar-dependent inputs, while the second term optimizes the target branch on unlabeled target samples under a token-augmented target-domain denoising context. The target branch is therefore trained to use visual-token conditioning on target-domain data, but it is not trained with target labels. During generation, the same cross-attention format is reused with source-image tokens and the source class prompt to synthesize labeled target-style samples. The token encoder is trained through the same denoising objective as the diffusion adapters, without additional alignment, reconstruction, or token-level supervision. The overall training and generation pipeline of VT-DUDA is illustrated in Fig. 1.

### 3.4 Generation-based UDA with Token-conditioned Synthesis

After training the token-conditioned diffusion model, we construct a synthetic labeled target-style dataset

$$\widetilde{\mathcal{D}}_{\mathrm{t}} = \{(\tilde{x}_k^{\mathrm{t}}, \tilde{y}_k)\}_{k=1}^{N_{\mathrm{syn}}}. \tag{14}$$

This synthetic dataset is constructed by assigning source labels to target-style samples generated from source-conditioned inputs.

**Target-style synthesis from Gaussian noise.** Let $z_T \sim \mathcal{N}(0, I)$ denote the initial latent noise, and let $\mathcal{G}(z_T; S, \sigma)$ denote a fixed diffusion sampler that maps an initial noisy latent, a conditioning context $S$, and a domain indicator $\sigma$ to a final latent prediction $\tilde{z}_0$. To synthesize a labeled target-style sample, we first draw a labeled source instance $(x_i^{\mathrm{s}}, y_i^{\mathrm{s}}) \sim \mathcal{D}_{\mathrm{s}}$, extract its visual tokens $U_i^{\mathrm{s}} = G_\psi(x_i^{\mathrm{s}})$, and construct the target-style conditioning context with a generation-time token-strength coefficient $\eta_{\mathrm{gen}} > 0$:

$$S^{\mathrm{t}}(y_i^{\mathrm{s}}; i) = \Big[ e\big(p_{\mathrm{s}}(y_i^{\mathrm{s}})\big) ; \eta_{\mathrm{gen}} U_i^{\mathrm{s}} \Big]. \tag{15}$$

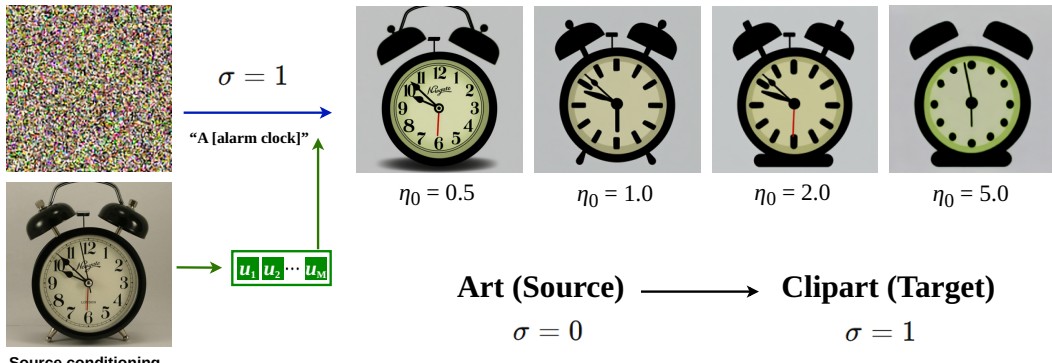

Figure 2: Token-strength scaling for target-style synthesis (Art→Clipart). For each labeled source instance, we extract visual tokens, combine them with the class prompt, and generate a target-style sample under the target branch. The figure illustrates the effect of varying the token strength $\eta \in \{0.5, 1.0, 2.0, 5.0\}$ on the generated samples.

Unless otherwise stated, we use $\eta_{\text{gen}} = \kappa_{\text{t}}$, and generate a target-style latent under the target branch using this conditioning context:

$$\tilde{z}_0^{\text{t}} = \mathcal{G}\big(z_T; S^{\text{t}}(y_i^{\text{s}}; i), \sigma = 1\big), \qquad z_T \sim \mathcal{N}(0, I), \tag{16}$$

and decode it into image space as $\tilde{x}^{\text{t}} = D_{\text{VAE}}(\tilde{z}_0^{\text{t}})$.

We assign the synthetic label $\tilde{y}^{\text{t}} = y_i^{\text{s}}$ by inheriting the source label associated with the conditioning source image, as in prior generation-based UDA pipelines. We do not assume this assignment is noise-free; instead, we evaluate it post hoc through the label-consistency diagnostic in Appendix A.7.2.

**Downstream UDA training.** Let $f_\phi$ denote the downstream classifier, with class posterior $p_\phi(y \mid x)$. Once the synthetic dataset $\widetilde{\mathcal{D}}_{\text{t}}$ has been constructed, we train $f_\phi$ using supervised learning on $\widetilde{\mathcal{D}}_{\text{t}}$, optionally combined with an unsupervised target regularizer on $\mathcal{D}_{\text{t}}$:

$$\mathcal{L}_{\text{uda}}(\phi) = \mathcal{L}_{\text{sup}}\big(\widetilde{\mathcal{D}}_{\text{t}}; \phi\big) + \lambda_{\text{u}} \, \mathcal{L}_{\text{unsup}}\big(\mathcal{D}_{\text{t}}; \phi\big), \tag{17}$$

where

$$\mathcal{L}_{\text{sup}}\big(\widetilde{\mathcal{D}}_{\text{t}}; \phi\big) = \mathbb{E}_{(\tilde{x}, \tilde{y}) \sim \widetilde{\mathcal{D}}_{\text{t}}}\Big[ -\log p_\phi(\tilde{y} \mid \tilde{x}) \Big]. \tag{18}$$

The unsupervised term $\mathcal{L}_{\text{unsup}}$ is instantiated by the chosen downstream UDA method, e.g., MCC or ELS in our experiments. VT-DUDA therefore changes the synthetic data construction stage while remaining compatible with different classifier-side UDA objectives.

**Translation-based augmentation.** The same visual-token interface is compatible with inversion-based translation procedures. Let $z_{i,0}^{\text{s}} = E_{\text{VAE}}(x_i^{\text{s}})$ be the source latent, and let $\mathcal{I}$ denote an inversion operator that maps this latent to an approximate terminal latent

$$z_{i,T}^{\text{inv}} = \mathcal{I}(z_{i,0}^{\text{s}}). \tag{19}$$

A translated target-style sample can then be obtained by running the sampler under the target branch using the same generation-time conditioning context $S^{\text{t}}(y_i^{\text{s}}; i)$:

$$\tilde{z}_{i,0}^{\text{inv}\to\text{t}} = \mathcal{G}\big(z_{i,T}^{\text{inv}}; S^{\text{t}}(y_i^{\text{s}}; i), \sigma = 1\big), \qquad \tilde{x}_i^{\text{inv}\to\text{t}} = D_{\text{VAE}}\big(\tilde{z}_{i,0}^{\text{inv}\to\text{t}}\big). \tag{20}$$

Such translated samples can be incorporated as an additional source of synthetic training data. In VT-DUDA, however, the conditioning mechanism itself remains unchanged: both Gaussian-noise synthesis and translation-based augmentation are driven by the same token-conditioned interface.

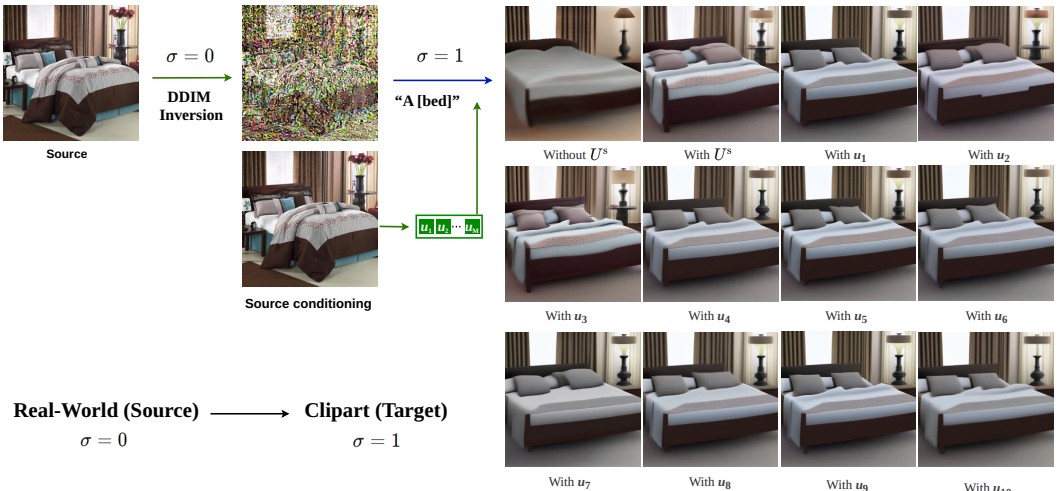

Figure 3: Token-subset manipulation under translation-based augmentation (Real-World→Clipart). Starting from a labeled source image in the Real-World domain, we first obtain an inverted latent and then re-synthesize it under the target branch using different token subsets. The figure shows how the tokenized conditioning interface allows explicit modification of the visual-token context during target-style translation.

### 3.5 Inference-time Token Manipulation

Because visual guidance is represented as an explicit token sequence, the conditioning context can be modified directly at inference time without retraining. Given a prompt $p$ and a reference image $v$, we extract visual tokens

$$U = G_\psi(v) \in \mathbb{R}^{M \times d_c}. \tag{21}$$

Let $\mathcal{M}_{\mathrm{tok}} \subseteq \{1, \ldots, M\}$ be a selected token index set and let $\eta > 0$ be a token-strength coefficient. We define the manipulated conditioning context

$$S_{\mathrm{gen}} = \big[ e(p) \,;\, \eta \, U_{\mathcal{M}_{\mathrm{tok}}} \big], \qquad U_{\mathcal{M}_{\mathrm{tok}}} \in \mathbb{R}^{|\mathcal{M}_{\mathrm{tok}}| \times d_c}. \tag{22}$$

These operations require no retraining and preserve the original diffusion backbone and adapter parameters. They do not assume that individual visual tokens are semantically disentangled or directly interpretable. Rather, they provide a direct way to modify the composition and scale of the visual-token context passed to cross-attention. Figure 2 visualizes token-strength scaling under target-style synthesis, while Fig. 3 illustrates token-subset manipulation in the translation-based setting.

## 4 Experiments

### 4.1 Experimental Setup

We evaluate VT-DUDA on three standard UDA benchmarks: Office-31 Saenko et al. (2010), Office-Home Venkateswara et al. (2017), and VisDA-2017 Peng et al. (2017). Office-31 contains 31 categories across three domains: Amazon (A), Webcam (W), and DSLR (D). Office-Home contains 65 categories across four domains: Art (Ar), Clipart (Cl), Product (Pr), and Real-World (Rw). VisDA-2017 contains a large synthetic-to-real shift with 12 categories. We follow the standard splits and report target-domain classification accuracy (%).

We compare VT-DUDA with representative discriminative UDA baselines and diffusion-based data-augmentation baselines. In particular, we instantiate VT-DUDA on top of MCC Jin et al. (2020) and ELS Zhang et al. (2023b), and compare against both the original discriminative methods and diffusion-based baselines including Terra Zhuang et al. (2024) and DCDM Zhang et al. (2025). For VT-DUDA, the

*full configuration* combines pure-noise target-style synthesis with inversion-based translation. Importantly, in our method these are two data-construction routes under the same token-conditioned generation framework: both use the proposed visual-token interface, and the translation route also conditions on visual tokens extracted from the source image being inverted. We additionally report an *inversion-free* protocol in which the synthetic labeled training set is constructed only from pure-noise target-style generation. This lighter setting is used to examine how far the proposed conditioning mechanism remains effective without the extra translation route.

Unless otherwise stated, VT-DUDA generates 50 images per class on Office-31 and Office-Home, and 1000 images per class on VisDA-2017. For Terra, we follow the same synthetic-data budget in our comparisons. For DCDM, we report the results under its published generation setting, which uses a larger synthetic-data budget: 200 generated images per class on Office-31 and Office-Home, and 2000 generated images per class on VisDA-2017. The downstream classifier uses ResNet-50 on Office-31/Office-Home and ResNet-101 on VisDA-2017. The image-to-token encoder $G_\psi$ is implemented with a ResNet-18 backbone He et al. (2016). We set the token count to $M = 10$ in the default configuration. All results are averaged over three runs.

Our empirical objective is practical: we evaluate whether the proposed conditioning design improves the downstream usefulness of synthetic target-style data for UDA. We therefore treat the target-domain classification accuracy obtained from the constructed synthetic set as the primary criterion throughout this section. Additional implementation details, evaluation protocols, and supplementary results are provided in the Appendix A.

## 4.2   Main Results

Tables 1 and 2 summarize the main UDA results under the full configuration described above. Across all three benchmarks, VT-DUDA achieves the strongest average performance when instantiated on top of MCC and ELS.

A first observation is that VT-DUDA consistently improves over the underlying discriminative UDA baselines. Relative to MCC, VT-DUDA improves the average accuracy by +4.01% on Office-Home (76.25 vs. 72.24), +1.92% on Office-31 (91.53 vs. 89.61), and +4.16% on VisDA-2017 (87.48 vs. 83.32). Relative to ELS, the gains are +4.90% on Office-Home (76.74 vs. 71.84), +1.83% on Office-31 (92.04 vs. 90.21), and +5.02% on VisDA-2017 (88.42 vs. 83.40). These margins indicate that the proposed token-conditioned synthetic-data construction substantially strengthens both downstream UDA objectives.

A second observation is that VT-DUDA also improves over strong diffusion-based data-augmentation baselines. Under the full setting, the best non-VT diffusion results are 76.26 on Office-Home (ELS+Terra), 91.80 on Office-31 (ELS+DCDM), and 86.86 on VisDA-2017 (ELS+Terra). The corresponding best VT-DUDA results are 76.74, 92.04, and 88.42, yielding gains of +0.48%, +0.24%, and +1.56%, respectively. This shows that the proposed conditioning design remains effective in the strongest practical generation setting and yields improvements over existing diffusion-based augmentation pipelines across all three benchmarks.

These full-setting results should be interpreted as the performance of the complete VT-DUDA pipeline. In our method, pure-noise synthesis and inversion-based translation are not two unrelated components; both are driven by the same visual-token-conditioned interface. The main results therefore evaluate VT-DUDA in its intended full form. To further examine the role of the conditioning design under a lighter generation protocol, we report an inversion-free variant in Sec. 4.3.

## 4.3   Inversion-free VT-DUDA and Budget-aware Comparison

Table 3 reports the inversion-free variant of VT-DUDA, where the synthetic labeled training set is constructed only from pure-noise target-style generation. This setting is useful for two reasons. First, it shows whether the proposed token-conditioned interface remains effective without the extra translation route. Second, it provides a direct way to examine data efficiency, since VT-DUDA uses only 50 generated images per class on Office-31 and Office-Home, and 1000 on VisDA-2017.

Table 1: Transfer accuracies (%) on Office-Home under the UDA setting. Results are averaged over 12 transfer tasks (Avg). The best result in each column is highlighted in **bold**.

| Method | Ar→Cl | Ar→Pr | Ar→Rw | Cl→Ar | Cl→Pr | Cl→Rw | Pr→Ar | Pr→Cl | Pr→Rw | Rw→Ar | Rw→Cl | Rw→Pr | Avg |
|---|---|---|---|---|---|---|---|---|---|---|---|---|---|
| ERM Vapnik (2013) | 44.06 | 67.12 | 74.26 | 53.26 | 61.96 | 64.54 | 51.91 | 38.90 | 72.94 | 64.51 | 43.84 | 75.39 | 59.39 |
| DANN Ganin et al. (2016) | 52.53 | 62.57 | 73.20 | 56.89 | 67.02 | 68.34 | 58.37 | 54.14 | 78.31 | 70.78 | 60.76 | 80.57 | 65.29 |
| CDAN Long et al. (2018) | 54.21 | 72.18 | 78.29 | 61.97 | 71.43 | 72.39 | 62.96 | 55.68 | 80.68 | 74.71 | 61.22 | 83.68 | 69.12 |
| AFN Xu et al. (2019) | 52.58 | 72.42 | 76.96 | 64.90 | 71.14 | 72.91 | 64.08 | 51.29 | 77.83 | 72.21 | 57.46 | 82.09 | 67.99 |
| MDD Zhang et al. (2019) | 56.37 | 75.53 | 79.17 | 62.95 | 73.21 | 73.55 | 62.56 | 54.86 | 79.49 | 73.84 | 61.45 | 84.06 | 69.75 |
| SDAT Rangwani et al. (2022) | 58.20 | 77.46 | 81.35 | 66.06 | 76.45 | 76.41 | 63.70 | 56.69 | 82.49 | 76.02 | 62.09 | 85.24 | 71.85 |
| MSGD Xia et al. (2022) | 58.70 | 76.90 | 78.90 | 70.10 | 76.20 | 76.60 | 69.00 | 57.20 | 82.30 | 74.90 | 62.70 | 84.50 | 72.40 |
| MCC Jin et al. (2020) | 56.83 | 79.81 | 82.66 | 67.80 | 77.02 | 77.82 | 66.98 | 55.43 | 81.79 | 73.95 | 61.41 | 85.44 | 72.24 |
| ELS Zhang et al. (2023b) | 57.79 | 77.65 | 81.62 | 66.59 | 76.74 | 76.43 | 62.69 | 56.69 | 82.12 | 75.63 | 62.85 | 85.35 | 71.84 |
| **Diffusion-based methods** | | | | | | | | | | | | | |
| MCC+Terra Zhuang et al. (2024) | 63.49 | 81.51 | 83.46 | 72.52 | 82.89 | 81.25 | 73.20 | 61.66 | 83.16 | 74.36 | 63.45 | 84.41 | 75.45 |
| ELS+Terra Zhuang et al. (2024) | 64.62 | 82.33 | 83.60 | 71.19 | 84.25 | 80.31 | 73.00 | 63.57 | 83.81 | 76.20 | **66.56** | 85.70 | 76.26 |
| MCC+DCDM Zhang et al. (2025) | 58.23 | 80.33 | 82.91 | 70.14 | 79.15 | 81.36 | 68.49 | 57.75 | 83.44 | 74.18 | 63.81 | 85.67 | 73.71 |
| ELS+DCDM Zhang et al. (2025) | 60.35 | 78.81 | 82.74 | 69.59 | 80.53 | 79.55 | 65.16 | 58.26 | 83.11 | 75.81 | 64.18 | 85.55 | 73.64 |
| MCC+VT-DUDA | **65.49** | 80.89 | **85.36** | 74.38 | 81.88 | **83.01** | **75.06** | 61.94 | 82.10 | 76.25 | 65.30 | 83.30 | 76.25 |
| ELS+VT-DUDA | 63.26 | **83.74** | 85.03 | 72.63 | **85.67** | 78.89 | 71.55 | **64.91** | **85.29** | **77.58** | 65.17 | **87.16** | **76.74** |

Table 2: Transfer accuracies (%) on Office-31 under the UDA setting, with VisDA-2017 mean accuracy (%) reported in the last column. The best result in each column is highlighted in **bold**.

| Dataset | Office-31 | | | | | | | VisDA-2017 |
|---|---|---|---|---|---|---|---|---|
| Method | A→W | D→W | W→D | A→D | D→A | W→A | Avg | mean |
| ERM Vapnik (2013) | 77.07 | 96.60 | 99.20 | 81.08 | 64.11 | 64.01 | 80.35 | 51.47 |
| DANN Ganin et al. (2016) | 89.85 | 97.95 | 99.90 | 83.26 | 73.28 | 73.75 | 86.33 | 79.02 |
| CDAN Long et al. (2018) | 92.42 | 98.62 | **100.00** | 91.44 | 74.61 | 72.80 | 88.32 | 80.74 |
| AFN Xu et al. (2019) | 91.82 | 98.77 | **100.00** | 95.12 | 72.43 | 70.71 | 88.14 | 74.64 |
| MDD Zhang et al. (2019) | 93.55 | 98.66 | **100.00** | 93.92 | 75.29 | 73.95 | 89.23 | 81.10 |
| SDAT Rangwani et al. (2022) | 91.32 | 98.83 | **100.00** | 95.25 | 76.97 | 73.19 | 89.26 | 83.23 |
| MSGD Xia et al. (2022) | 95.50 | 99.20 | **100.00** | 95.60 | 77.30 | 77.00 | 90.80 | 84.60 |
| MCC Jin et al. (2020) | 94.09 | 98.32 | 99.67 | 94.25 | 75.89 | 75.46 | 89.61 | 83.32 |
| ELS Zhang et al. (2023b) | 93.84 | 98.78 | **100.00** | 95.78 | 77.72 | 75.13 | 90.21 | 83.40 |
| **Diffusion-based methods** | | | | | | | | |
| MCC+Terra Zhuang et al. (2024) | 94.55 | 99.03 | **100.00** | 96.46 | 78.64 | 79.37 | 91.34 | 85.39 |
| ELS+Terra Zhuang et al. (2024) | 94.09 | **99.21** | **100.00** | 96.25 | 78.67 | 79.45 | 91.28 | 86.86 |
| MCC+DCDM Zhang et al. (2025) | 95.51 | 98.58 | 99.93 | 95.31 | 78.26 | 78.43 | 91.01 | 86.56 |
| ELS+DCDM Zhang et al. (2025) | 96.90 | 98.91 | **100.00** | 97.46 | **79.79** | 77.74 | 91.80 | 86.29 |
| MCC+VT-DUDA | 96.40 | 98.98 | **100.00** | 98.34 | 77.33 | 78.13 | 91.53 | 87.48 |
| ELS+VT-DUDA | **98.36** | 95.97 | **100.00** | **99.46** | 78.11 | **80.34** | **92.04** | **88.42** |

Even under this lighter protocol, VT-DUDA remains consistently stronger than the underlying discriminative UDA baselines. Under MCC, the gains over the original discriminative baseline are +1.26% on Office-Home, +1.76% on Office-31, and +3.60% on VisDA-2017. Under ELS, the corresponding gains are +1.88%, +1.71%, and +4.16%.

Compared with Terra under the same pure-noise budget, VT-DUDA also performs better on Office-Home. Under MCC, VT-DUDA improves over Terra by +1.66%, while under ELS the corresponding gain is +1.54%. Because Terra and VT-DUDA use the same generation budget in this comparison, this result provides cleaner evidence for the benefit of visual-token conditioning.

The budget-aware comparison is particularly revealing on Office-31 and VisDA-2017. On these two benchmarks, inversion-free VT-DUDA already surpasses the full Terra configuration, even though the latter additionally uses inversion-based translation. Specifically, on Office-31, MCC+VT-DUDA reaches 91.37% versus 91.34% for MCC+Terra, and ELS+VT-DUDA reaches 91.92% versus 91.28% for ELS+Terra. On VisDA-2017, MCC+VT-DUDA reaches 86.92% versus 85.39% for MCC+Terra, and ELS+VT-DUDA reaches

Table 3: Average accuracies (%) under the inversion-free protocol. VT-DUDA and Terra use only pure-noise target-style generation, with 50 generated images per class on Office-31 and Office-Home and 1000 on VisDA-2017. DCDM is included for reference under its published larger-budget setting (200 per class on Office-31/Office-Home and 2000 on VisDA-2017).

| Method | Office-Home Avg | Office-31 Avg | VisDA mean |
|---|---|---|---|
| MCC Jin et al. (2020) | 72.24 | 89.61 | 83.32 |
| ELS Zhang et al. (2023b) | 71.84 | 90.21 | 83.40 |
| MCC+Terra Zhuang et al. (2024) | 71.84 | 91.34 | 85.39 |
| ELS+Terra Zhuang et al. (2024) | 72.18 | 91.28 | 86.86 |
| MCC+DCDM Zhang et al. (2025) | 73.71 | 91.01 | 86.56 |
| ELS+DCDM Zhang et al. (2025) | 73.64 | 91.80 | 86.29 |
| MCC+VT-DUDA | 73.50 | 91.37 | 86.92 |
| ELS+VT-DUDA | **73.72** | **91.92** | **87.56** |

87.56% versus 86.86% for ELS+Terra. In other words, on these two benchmarks, using only 50 generated images per class on Office-31 and 1000 on VisDA-2017 is already sufficient for VT-DUDA to outperform a stronger Terra pipeline that additionally includes inversion-based translation.

A similarly strong pattern appears in the comparison with DCDM, despite DCDM using a much larger synthetic-data budget. On Office-31, inversion-free VT-DUDA outperforms DCDM under both downstream objectives (91.37% vs. 91.01% for MCC and 91.92% vs. 91.80% for ELS) while using 50, rather than 200, generated images per class. On VisDA-2017, VT-DUDA again exceeds DCDM under both objectives (86.92% vs. 86.56% for MCC and 87.56% vs. 86.29% for ELS) while using 1000, rather than 2000, generated images per class. On Office-Home, VT-DUDA remains competitive with DCDM under this smaller budget: ELS+VT-DUDA slightly exceeds ELS+DCDM (73.72% vs. 73.64%), while MCC+VT-DUDA is within 0.21 points of MCC+DCDM (73.50% vs. 73.71%). To visualize how the generated target-style samples relate to the target domain, we present t-SNE plots in Fig. 4. The synthesized samples occupy a feature region that is closer to the unlabeled target domain than the original source-domain samples. Taken together, these results indicate that the proposed token-conditioned interface improves not only final accuracy but also the downstream utility per generated sample.

### 4.4 Inversion-based Translation within the Same Token-conditioned Framework

The comparison between Tables 1, 2, and 3 shows that inversion-based translation further improves the strongest practical configuration. In VT-DUDA, inversion-based translation is an additional synthetic-data construction route driven by the same visual-token-conditioned generation framework. From this perspective, the inversion-free results and the full results answer two different questions. The inversion-free results show that the proposed conditioning interface is already effective under a lighter pure-noise protocol, while the full results show that it remains strong when combined with an additional inversion-based augmentation route. We therefore view inversion-based translation as complementary to token-conditioned synthesis: it can further strengthen the complete pipeline.

### 4.5 Ablation Studies

We conduct ablations under the inversion-free protocol so that the observed trends can be attributed directly to the token-conditioned generator without adding the extra translation route.

Table 4 shows two representative trends. First, increasing the number of generated images per class consistently improves downstream performance. This is expected because the downstream classifier is trained on synthetic target-style data, and a larger synthetic set provides more supervision. Second, varying the number of visual tokens $M$ shows that a compact token set already works well, whereas overly long token sequences degrade performance. In our experiments, $M = 5$ or $M = 10$ is sufficient to obtain the strongest results, while increasing the token count to 20 or 50 hurts accuracy. We present this as an empirical observation

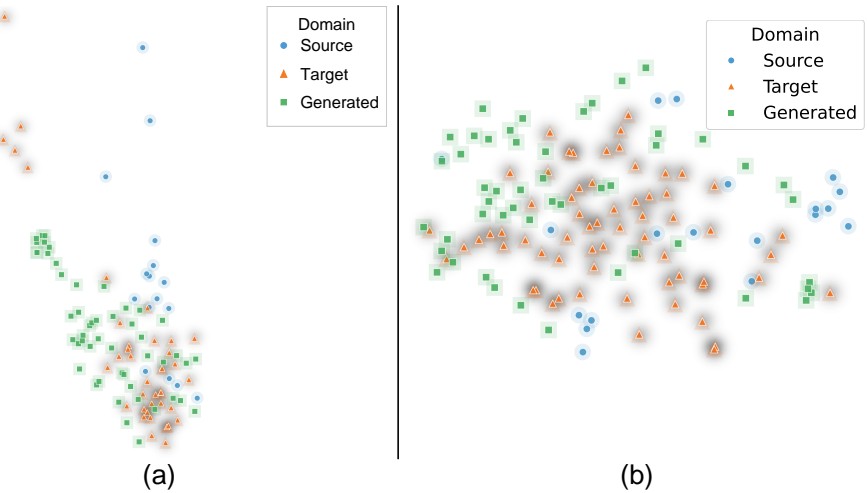

Figure 4: t-SNE visualization on Office-Home under the UDA setting. We compare source-domain features, target-domain features, and features from the generated target-style domain. (a) Ar→Cl for class *Webcam*. (b) Ar→Rw for class *Toys*.

Table 4: Ablation results on Office-Home (% accuracy).

| Ablation setting | Value | MCC+VT-DUDA | ELS+VT-DUDA |
|---|---|---|---|
| *Ar→Cl: number of generated images per class* | | | |
| Generated images / class | 10 | 40.18 | 47.74 |
| Generated images / class | 25 | 50.54 | 52.83 |
| Generated images / class | 50 | 57.77 | 59.24 |
| Generated images / class | 100 | **61.15** | **63.46** |
| *Cl→Rw: number of visual tokens M (50 generated images / class)* | | | |
| Visual tokens $M$ | 5 | 82.16 | **83.44** |
| Visual tokens $M$ | 10 | **82.59** | 83.30 |
| Visual tokens $M$ | 20 | 81.34 | 82.53 |
| Visual tokens $M$ | 50 | 79.32 | 80.10 |

about conditioning efficiency. Appendix A.4 further provides detailed transfer-wise results on Office-Home and Office-31, together with per-class results on VisDA-2017, under the inversion-free protocol.

## 5 Limitations and Broader Impact

VT-DUDA is evaluated in the standard image-classification UDA setting, and its main empirical claim is limited to the downstream usefulness of generated target-style data on Office-31, Office-Home, and VisDA-2017. The visual-token sequence is intended to provide compact instance-level conditioning, but it should not be interpreted as a dense spatial-control signal or as a guarantee that individual tokens correspond to human-interpretable semantic parts. As shown by the post-hoc label-consistency analysis in Appendix A.7.2, generated samples may preserve class information to different degrees across transfer directions, so deployment-oriented use should validate the final classifier on task-relevant real target data whenever possible.

Because VT-DUDA relies on image-conditioned generation, synthetic samples may inherit source-domain biases or introduce class-irrelevant correlations that affect downstream classifiers. In addition, image-conditioned generation mechanisms can have dual-use risks if repurposed outside the academic UDA setting considered here. VT-DUDA is not designed for identity forgery or person-specific generation, but such risks are relevant to image-generation systems more broadly. Finally, the SDXL-based pipeline has non-trivial computational cost; Appendix A.8 reports measured training time, memory use, and generation latency to make this accuracy–cost trade-off explicit.

## 6 Conclusion

We introduced VT-DUDA, a visual-token conditioning framework for diffusion-guided unsupervised domain adaptation. The method augments the cross-attention context of an adapter-based latent diffusion model with instance-level visual tokens, enriching text-based conditioning without modifying the diffusion backbone or objective. Experiments on Office-31, Office-Home, and VisDA-2017 show that this design improves the downstream utility of synthetic target-style data over strong baselines, including under limited generation budgets. These results suggest that conditioning design matters in generation-based UDA. Evaluating this approach beyond image classification and the diffusion setting considered here remains future work.

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

# A    Appendix

## A.1    Implementation Details

We instantiate VT-DUDA on top of Stable Diffusion XL (SDXL) Podell et al. (2024). We use the SDXL VAE, keep the VAE frozen throughout training, and use the same VAE encoder $E_{\text{VAE}}$ and decoder $D_{\text{VAE}}$ as defined in Sec. 3.2. In the implementation, the standard SDXL latent scaling factor is applied when encoding images into latents and inverted when decoding generated latents back to images.

**Low-rank adaptation.**    We freeze the pretrained UNet and both SDXL text encoders, and optimize only lightweight adapter parameters together with the image-to-token encoder. In the UNet, LoRA modules are attached to all attention projection matrices, and the LoRA rank is set to $r = 32$. The branch-dependent updates in Eq. equation 5 are implemented as effective low-rank updates Zhuang et al. (2024) rather than as two independent LoRA modules. Specifically, for an adaptable weight matrix $W_0 \in \mathbb{R}^{m \times n}$, we use

$$\Delta W^{(\sigma)} = W_{\text{up}}\big(I + \sigma\Omega_{\text{lora}}\big)^{\top} W_{\text{down}}, \qquad \sigma \in \{0, 1\}, \tag{23}$$

where $W_{\text{up}} \in \mathbb{R}^{m \times r}$, $W_{\text{down}} \in \mathbb{R}^{r \times n}$, $I \in \mathbb{R}^{r \times r}$, and $\Omega_{\text{lora}} \in \mathbb{R}^{r \times r}$ is a trainable target-activated modulation matrix. Hence, $\Delta W^{(0)} = W_{\text{up}} W_{\text{down}}$ and $\Delta W^{(1)} = W_{\text{up}}(I + \Omega_{\text{lora}})^{\top} W_{\text{down}}$. This realizes the domain-dependent adapters in Eq. equation 5 through a shared low-rank basis $(W_{\text{up}}, W_{\text{down}})$ and a target-specific modulation. In practice, the source and target forward passes are executed sequentially within each iteration using the binary branch indicator $\sigma$. No additional LoRA scaling coefficient is used, i.e., the effective scaling in Eq. equation 2 is $\lambda = 1$ in our implementation.

**Image-to-token encoder.**    The image-to-token encoder $G_\psi$ in Equation 6 uses a **ResNet-18** backbone. Specifically, the ResNet-18 backbone extracts a global visual representation from an input image $v$, which is projected to an $M$-dimensional latent code $a(v) \in \mathbb{R}^M$ with $M = 10$. Each scalar component $a_m(v)$ is then mapped to a visual token $u_m \in \mathbb{R}^{d_c}$ through an independent token-specific MLP $g_m$, following Eq. equation 7. In our implementation, these token-specific heads use a $1 \rightarrow 64 \rightarrow 128 \rightarrow d_c$ multilayer perceptron with ELU activations. The cross-attention dimension $d_c$ is set automatically to the concatenated text-conditioning dimension of the two frozen SDXL text encoders. The token encoder is trained jointly with the diffusion adapters using only the denoising objective in equation 13.

**Prompt construction.**    Let $p_{\text{s}}(y)$ and $p_{\text{t}}$ denote the source and target prompts introduced in Sec. 3.3.2. In the implementation, the source prompt is instantiated from the source image folder name using a simple class template, e.g., `"A backpack"`, while the target prompt is the generic template $p_{\text{t}} = $ `"An image"`. Thus, source prompts are label-dependent, whereas target prompts are domain-generic.

**Fusion of text and visual conditioning.**    We follow the hybrid context formulation in Eq. equation 10. Text conditioning is obtained from the two frozen SDXL text encoders. Given a prompt $p$, each text encoder outputs a token sequence, and we use the penultimate hidden state from each encoder. These two token sequences are concatenated along the channel dimension to form the token-wise text embedding $R = e(p) \in \mathbb{R}^{M_p \times d_c}$. Visual tokens are appended only to this cross-attention token sequence.

In addition to the cross-attention token sequence, SDXL uses added conditioning variables such as pooled text embeddings and time-size embeddings. We keep these SDXL-specific added conditioning variables unchanged: the pooled text embedding is computed from the corresponding text prompt using the frozen SDXL text encoder, and the time-size embedding is computed from the original image size, target image size, and crop coordinates. In our experiments, the original size and target size are both $1024 \times 1024$, and the crop offset is fixed to $(0, 0)$. The visual tokens are not pooled and are not injected into the SDXL added-conditioning pathway; they only extend the cross-attention context passed as token-wise conditioning to the UNet.

For a source image $x_i^{\mathrm{s}}$ and a target image $x_j^{\mathrm{t}}$, we extract visual tokens $U_i^{\mathrm{s}} = G_\psi(x_i^{\mathrm{s}})$ and $U_j^{\mathrm{t}} = G_\psi(x_j^{\mathrm{t}})$, with $M = 10$ tokens per image. We then construct the cross-attention contexts

$$S_i^{\mathrm{s}} = \left[R_i^{\mathrm{s}}; \kappa_{\mathrm{s}} U_i^{\mathrm{s}}\right],$$
$$S_j^{\mathrm{t}} = \left[R_j^{\mathrm{t}}; \kappa_{\mathrm{t}} U_j^{\mathrm{t}}\right],$$

exactly as in Eq. equation 10. In our default setting, $\kappa_{\mathrm{s}} = \kappa_{\mathrm{t}} = 1$. No extra fusion block is added: text tokens and visual tokens are concatenated directly along the sequence dimension and passed to the UNet cross-attention layers.

**Training objective and batch composition.** We optimize the joint objective in Equation 13 with equal weights $\lambda_{\mathrm{s}} = \lambda_{\mathrm{t}} = 1$. Each optimization step processes one minibatch of paired source and target samples. Each iteration contains 8 source images and 8 target images. The source and target losses are computed in two sequential passes through the same frozen diffusion backbone with different branch indicators:

$$\mathcal{L}_{\mathrm{train}} = \mathcal{L}_{\mathrm{s}} + \mathcal{L}_{\mathrm{t}}.$$

where $\mathcal{L}_{\mathrm{s}}$ uses $\sigma = 0$ and context $S_i^{\mathrm{s}}$, and $\mathcal{L}_{\mathrm{t}}$ uses $\sigma = 1$ and context $S_j^{\mathrm{t}}$. For both passes, the image is encoded to the latent space with the frozen VAE, a diffusion timestep $\tau$ is sampled uniformly, and Gaussian noise $\epsilon \sim \mathcal{N}(0, I)$ is added using the pretrained DDPM scheduler, following Eq. equation 3. The UNet is trained to predict the added noise $\epsilon$, and we optimize the standard mean-squared denoising loss used in Eq. equation 13. Importantly, within each iteration, the same sampled $\tau$ and the same noise tensor $\epsilon$ are reused for the source and target passes, so the two losses are matched under identical corruption conditions.

**Optimization and training configuration.** We train on a single NVIDIA A100 (80GB) using mixed precision `fp16`. The VAE is always kept in `float32` for numerical stability, and the image-to-token encoder is also optimized in `float32`. The launch configuration uses 8-bit AdamW with $\beta_1 = 0.9$, $\beta_2 = 0.999$, $\epsilon = 10^{-8}$, and weight decay $10^{-4}$ for the UNet adapter parameters. The optimizer uses separate parameter groups: the low-rank factors $(W_{\mathrm{up}}, W_{\mathrm{down}})$ are optimized with learning rate $5 \times 10^{-5}$, while the domain-modulation parameters $\Omega$ are optimized with learning rate $5 \times 10^{-3}$, corresponding to `lr_theta`=100. The image-to-token encoder $G_\psi$ is optimized with learning rate $10^{-4}$ and weight decay $10^{-4}$. Only the UNet adapters and $G_\psi$ are updated.

We train for 8000 optimization steps with per-device batch size 8, gradient accumulation 1, and maximum gradient norm 1.0. The learning-rate scheduler is constant with zero warmup. Data loading uses 8 worker processes. The random seed is fixed to 0.

**Data preprocessing.** All training images are first corrected with EXIF transpose, converted to RGB if necessary, resized to $1024 \times 1024$ using bilinear interpolation, and then randomly cropped to $1024 \times 1024$. After cropping, each image is converted to a tensor and normalized channel-wise to $[-1, 1]$ using mean $(0.5, 0.5, 0.5)$ and standard deviation $(0.5, 0.5, 0.5)$. No horizontal flipping or additional color augmentation is applied in this training pipeline. The same normalized tensor is used both for VAE encoding and for visual-token extraction by $G_\psi$.

## A.2   Dataset and Evaluation Protocol Details

This appendix consolidates the benchmark setup, evaluation protocol, and the two synthetic-data construction routes used in VT-DUDA. We first describe the datasets and evaluation protocols in text form, and then give the complete algorithms for (i) pure-noise target-style synthesis and (ii) DDIM-inversion-based target-style translation.

### A.2.1   Benchmarks and evaluation protocol

We evaluate VT-DUDA on three standard unsupervised domain adaptation benchmarks.

**Office-31.** Office-31 contains three domains, Amazon (A), Webcam (W), and DSLR (D), with 31 shared object categories. Following the standard UDA protocol, we evaluate all six directed transfer tasks:

$$A{\to}W, \ D{\to}W, \ W{\to}D, \ A{\to}D, \ D{\to}A, \ W{\to}A.$$

For this benchmark, the downstream classifier uses a ResNet-50 backbone, and we report target-domain classification accuracy for each transfer task together with the average over all six tasks.

**Office-Home.** Office-Home contains four domains, Art (Ar), Clipart (Cl), Product (Pr), and Real-World (Rw), with 65 shared categories. We follow the standard protocol and evaluate all twelve directed transfer tasks formed by ordered pairs of distinct domains. The downstream classifier again uses a ResNet-50 backbone, and we report target-domain classification accuracy for each task together with the average over all twelve tasks.

**VisDA-2017.** VisDA-2017 contains a large synthetic-to-real domain shift with 12 categories. We follow the standard training/validation protocol, using the synthetic split as source and the real validation split as target. For this benchmark, the downstream classifier uses a ResNet-101 backbone, and we report the standard target-domain mean accuracy on the real validation split.

Across all benchmarks, the source and target domains share the same label space. Source labels are available only for the source domain, while target labels are not used during training. All reported numbers are averaged over three runs.

### A.2.2 Synthetic-data construction protocol

Let $(x_i^s, y_i^s) \in \mathcal{D}_s$ denote a labeled source sample, and let $U_i^s = G_\psi(x_i^s)$ be the corresponding visual-token sequence extracted by the image-to-token encoder. During target-style generation, VT-DUDA uses the target-side conditioning context

$$S_i^t = \Big[ e\big(p_s(y_i^s)\big) ; \, \eta_{\text{gen}} \, U_i^s \Big], \tag{24}$$

where $e(\cdot)$ is the text-embedding operator and $\eta_{\text{gen}} > 0$ is the generation-time token-strength coefficient. Unless otherwise stated, we use $\eta_{\text{gen}} = 1$.

VT-DUDA constructs synthetic target-style training data using two routes: (i) inversion-free pure-noise synthesis and (ii) DDIM-inversion-based target-style translation.

**Inversion-free protocol.** The inversion-free protocol uses only the pure-noise route. Unless otherwise stated, this protocol generates 50 images per class on Office-31 and Office-Home, and 1000 images per class on VisDA-2017. In this setting, the synthetic labeled target-style dataset is $\widetilde{\mathcal{D}}_t = \widetilde{\mathcal{D}}_t^{\text{noise}}$.

**Full protocol.** The full protocol augments the pure-noise route with the DDIM-inversion-based translation route. Both routes use the same visual-token-conditioned interface and assign the source label $y_i^s$ to the resulting target-style sample. Accordingly, the final synthetic labeled target-style dataset is

$$\widetilde{\mathcal{D}}_t = \widetilde{\mathcal{D}}_t^{\text{noise}} \ \cup \ \widetilde{\mathcal{D}}_t^{\text{inv}\to\text{t}}. \tag{25}$$

**Sampler and branch assignment.** In the pure-noise route, synthesis is performed under the target branch ($\sigma = 1$). Final image generation is instantiated with DPM-Solver multistep sampling using 25 inference steps. In the DDIM-based route, source images are first inverted under the source branch ($\sigma = 0$) and then translated under the target branch ($\sigma = 1$) on the same reduced timestep schedule.

### A.2.3 Downstream training and evaluation

After synthetic-data construction, we train a downstream target classifier on the resulting synthetic target-style dataset, optionally together with an unsupervised target regularizer, as described in the main paper. We instantiate this stage using MCC and ELS. The benchmark-specific backbones and evaluation metrics are exactly those described above.

---

**Algorithm 1** Pure-noise target-style synthesis with visual-token conditioning. The procedure is sampler-agnostic and is instantiated with DPM-Solver multistep sampling and $K = 25$ inference steps in our experiments.

---

**Require:** Labeled source dataset $\mathcal{D}_{\mathrm{s}}$, trained token-conditioned diffusion model $\epsilon_{\theta_0 + \Delta\theta}$, VAE decoder $D_{\mathrm{VAE}}$, source prompt constructor $p_{\mathrm{s}}(\cdot)$, image-to-token encoder $G_\psi$, class-wise generation budgets $\{B_c\}_{c=1}^C$, generation-time token strength $\eta_{\mathrm{gen}}$ (default: $\eta_{\mathrm{gen}} = 1$), inference schedule $\{\tau_k\}_{k=0}^K$, target-branch sampler updates $\Phi_{\tau_k \to \tau_{k-1}}$

**Ensure:** Synthetic labeled target-style dataset $\widetilde{\mathcal{D}}_{\mathrm{t}}^{\mathrm{noise}}$

1: Initialize $\widetilde{\mathcal{D}}_{\mathrm{t}}^{\mathrm{noise}} \leftarrow \varnothing$
2: **for** each class $c \in \{1, \dots, C\}$ **do**
3:     **for** $b = 1, \dots, B_c$ **do**
4:         Sample $(x_i^{\mathrm{s}}, y_i^{\mathrm{s}}) \sim \mathcal{D}_{\mathrm{s}}$ such that $y_i^{\mathrm{s}} = c$
5:         Extract source visual tokens $U_i^{\mathrm{s}} \leftarrow G_\psi(x_i^{\mathrm{s}})$
6:         Form target-style conditioning context $S_i^{\mathrm{t}} \leftarrow \left[ e\big(p_{\mathrm{s}}(y_i^{\mathrm{s}})\big) ; \eta_{\mathrm{gen}} U_i^{\mathrm{s}} \right]$
7:         Sample initial latent $z_{\tau_K} \sim \mathcal{N}(0, I)$
8:         **for** $k = K, K-1, \dots, 1$ **do**
9:             Predict target-branch noise $\hat{\epsilon}_k \leftarrow \epsilon_{\theta_0 + \Delta\theta}\big(z_{\tau_k}, \tau_k, S_i^{\mathrm{t}}, 1\big)$
10:            Update latent $z_{\tau_{k-1}} \leftarrow \Phi_{\tau_k \to \tau_{k-1}}\big(z_{\tau_k}, \hat{\epsilon}_k, S_i^{\mathrm{t}}, 1\big)$
11:         **end for**
12:         Decode $\tilde{x}^{\mathrm{t}} \leftarrow D_{\mathrm{VAE}}(z_{\tau_0})$
13:         Add $(\tilde{x}^{\mathrm{t}}, y_i^{\mathrm{s}})$ to $\widetilde{\mathcal{D}}_{\mathrm{t}}^{\mathrm{noise}}$
14:     **end for**
15: **end for**
16: **return** $\widetilde{\mathcal{D}}_{\mathrm{t}}^{\mathrm{noise}}$

---

### A.2.4   Pure-noise target-style synthesis with visual-token conditioning

The first synthetic-data route generates target-style samples directly from Gaussian noise under the target branch of the token-conditioned diffusion model. For a labeled source instance $(x_i^{\mathrm{s}}, y_i^{\mathrm{s}})$, we first extract the source visual tokens $U_i^{\mathrm{s}}$, build the target-style conditioning context $S_i^{\mathrm{t}}$ according to equation 24, and then run the diffusion sampler under the target branch ($\sigma = 1$) from an initial Gaussian latent.

To keep the formulation sampler-agnostic, let $\Phi_{\tau_k \to \tau_{k-1}}$ denote the update rule of the chosen diffusion sampler from timestep $\tau_k$ to $\tau_{k-1}$. In our experiments, final image generation is instantiated with DPM-Solver multistep sampling using 25 inference steps. Let $0 = \tau_0 < \tau_1 < \cdots < \tau_K = T$ denote the inference schedule. Starting from $z_{\tau_K} \sim \mathcal{N}(0, I)$, the target-branch denoising process iterates

$$\hat{\epsilon}_k = \epsilon_{\theta_0 + \Delta\theta}\big(z_{\tau_k}, \tau_k, S_i^{\mathrm{t}}, \sigma = 1\big), \tag{26}$$

$$z_{\tau_{k-1}} = \Phi_{\tau_k \to \tau_{k-1}}\big(z_{\tau_k}, \hat{\epsilon}_k, S_i^{\mathrm{t}}, \sigma = 1\big), \qquad k = K, \dots, 1. \tag{27}$$

The final latent $z_{\tau_0}$ is decoded by the VAE decoder to obtain the synthetic target-style image $\tilde{x}^{\mathrm{t}}$, and the label is inherited from the conditioning source image: $\tilde{y}^{\mathrm{t}} = y_i^{\mathrm{s}}$.

### A.2.5   DDIM inversion and target-style translation

The second synthetic-data route starts from a source image rather than Gaussian noise. Given a labeled source sample $(x_i^{\mathrm{s}}, y_i^{\mathrm{s}})$, we first encode it into the latent space, $z_{i,0}^{\mathrm{s}} = E_{\mathrm{VAE}}(x_i^{\mathrm{s}})$, and compute the source visual tokens $U_i^{\mathrm{s}} = G_\psi(x_i^{\mathrm{s}})$. We then form two conditioning contexts:

$$S_i^{\mathrm{s}} = \left[ e\big(p_{\mathrm{s}}(y_i^{\mathrm{s}})\big) ; \kappa_{\mathrm{s}} U_i^{\mathrm{s}} \right], \tag{28}$$

$$S_i^{\mathrm{t}} = \left[ e\big(p_{\mathrm{s}}(y_i^{\mathrm{s}})\big) ; \eta_{\mathrm{gen}} U_i^{\mathrm{s}} \right]. \tag{29}$$

The first is used during DDIM inversion under the source branch ($\sigma = 0$), and the second is used during target-style resynthesis under the target branch ($\sigma = 1$).

We use deterministic DDIM inversion on a reduced inference schedule $0 = \tau_0 < \tau_1 < \cdots < \tau_K = T$, where $\tau_0 = 0$ denotes the clean latent endpoint and $\bar{\alpha}_{\tau_0} = \bar{\alpha}_0 = 1$. The diffusion model is trained with timesteps sampled from $\{1, \ldots, T\}$. The endpoint $\tau_0 = 0$ is introduced as a notational boundary for the clean latent state, and its treatment follows the boundary convention of the chosen DDIM scheduler. At inversion step $k$, given the current latent $z^{\text{inv}}_{\tau_{k-1}}$, we predict

$$\hat{\epsilon}^{\text{s}}_k = \epsilon_{\theta_0 + \Delta\theta}\big(z^{\text{inv}}_{\tau_{k-1}}, \tau_{k-1}, S^{\text{s}}_i, \sigma = 0\big), \tag{30}$$

estimate the clean latent

$$\hat{z}^{\text{s}}_{0,k-1} = \frac{z^{\text{inv}}_{\tau_{k-1}} - \sqrt{1 - \bar{\alpha}_{\tau_{k-1}}}\, \hat{\epsilon}^{\text{s}}_k}{\sqrt{\bar{\alpha}_{\tau_{k-1}}}}, \tag{31}$$

and move to the next noisier state

$$z^{\text{inv}}_{\tau_k} = \sqrt{\bar{\alpha}_{\tau_k}}\, \hat{z}^{\text{s}}_{0,k-1} + \sqrt{1 - \bar{\alpha}_{\tau_k}}\, \hat{\epsilon}^{\text{s}}_k. \tag{32}$$

After $K$ steps, we obtain the approximate terminal latent $z^{\text{inv}}_{\tau_K}$.

Target-style translation then runs deterministic DDIM sampling from this inverted latent under the target branch. At translation step $k$, we predict

$$\hat{\epsilon}^{\text{t}}_k = \epsilon_{\theta_0 + \Delta\theta}\big(z^{\text{t}}_{\tau_k}, \tau_k, S^{\text{t}}_i, \sigma = 1\big), \tag{33}$$

estimate the clean latent

$$\hat{z}^{\text{t}}_{0,k} = \frac{z^{\text{t}}_{\tau_k} - \sqrt{1 - \bar{\alpha}_{\tau_k}}\, \hat{\epsilon}^{\text{t}}_k}{\sqrt{\bar{\alpha}_{\tau_k}}}, \tag{34}$$

and update

$$z^{\text{t}}_{\tau_{k-1}} = \sqrt{\bar{\alpha}_{\tau_{k-1}}}\, \hat{z}^{\text{t}}_{0,k} + \sqrt{1 - \bar{\alpha}_{\tau_{k-1}}}\, \hat{\epsilon}^{\text{t}}_k, \qquad k = K, \ldots, 1. \tag{35}$$

The final latent $z^{\text{t}}_{\tau_0}$ is decoded into the translated target-style image $\tilde{x}^{\text{inv}\to\text{t}}_i$, to which we assign the label $y^{\text{s}}_i$.

### A.2.6 Remarks on protocol consistency

Both synthetic-data routes are driven by the same token-conditioned interface. In the pure-noise route, the source label and source visual tokens provide the class and instance information used to synthesize a target-style sample from Gaussian noise. In the DDIM-based route, the same source visual tokens are used first to invert the source image under the source branch and then to resynthesize it under the target branch. This design keeps the source-conditioned signal consistent across the two synthetic-data construction routes while allowing the final sample to be rendered in the target style.

---

**Algorithm 2** DDIM-inversion-based target-style translation with visual-token conditioning.

---

**Require:** Selected labeled source subset $\mathcal{S}_{\mathrm{inv}} \subseteq \mathcal{D}_{\mathrm{s}}$, trained token-conditioned diffusion model $\epsilon_{\theta_0 + \Delta\theta}$, VAE encoder $E_{\mathrm{VAE}}$, VAE decoder $D_{\mathrm{VAE}}$, source prompt constructor $p_{\mathrm{s}}(\cdot)$, image-to-token encoder $G_{\psi}$, source-side token strength $\kappa_{\mathrm{s}}$, generation-time token strength $\eta_{\mathrm{gen}}$ (default: $\eta_{\mathrm{gen}} = 1$), reduced DDIM schedule $0 = \tau_0 < \tau_1 < \cdots < \tau_K = T$ with $\bar{\alpha}_{\tau_0} = 1$

**Ensure:** Synthetic labeled translated dataset $\widetilde{\mathcal{D}}_{\mathrm{t}}^{\mathrm{inv}\to\mathrm{t}}$

1: Initialize $\widetilde{\mathcal{D}}_{\mathrm{t}}^{\mathrm{inv}\to\mathrm{t}} \leftarrow \varnothing$
2: **for** each $(x_i^{\mathrm{s}}, y_i^{\mathrm{s}}) \in \mathcal{S}_{\mathrm{inv}}$ **do**
3:      Encode source image to latent $z_{i,0}^{\mathrm{s}} \leftarrow E_{\mathrm{VAE}}(x_i^{\mathrm{s}})$
4:      Extract source visual tokens $U_i^{\mathrm{s}} \leftarrow G_{\psi}(x_i^{\mathrm{s}})$
5:      Form source-side inversion context $S_i^{\mathrm{s}} \leftarrow \left[ e\big(p_{\mathrm{s}}(y_i^{\mathrm{s}})\big) ; \kappa_{\mathrm{s}} U_i^{\mathrm{s}} \right]$
6:      Form target-side resynthesis context $S_i^{\mathrm{t}} \leftarrow \left[ e\big(p_{\mathrm{s}}(y_i^{\mathrm{s}})\big) ; \eta_{\mathrm{gen}} U_i^{\mathrm{s}} \right]$
7:      Initialize $z_{\tau_0}^{\mathrm{inv}} \leftarrow z_{i,0}^{\mathrm{s}}$
8:      **for** $k = 1, \ldots, K$ **do**                  ▷ Deterministic DDIM inversion under source branch
9:          $\hat{\epsilon}_k^{\mathrm{s}} \leftarrow \epsilon_{\theta_0 + \Delta\theta}\big(z_{\tau_{k-1}}^{\mathrm{inv}}, \tau_{k-1}, S_i^{\mathrm{s}}, 0\big)$
10:         $\hat{z}_{0,k-1}^{\mathrm{s}} \leftarrow \dfrac{z_{\tau_{k-1}}^{\mathrm{inv}} - \sqrt{1 - \bar{\alpha}_{\tau_{k-1}}}\, \hat{\epsilon}_k^{\mathrm{s}}}{\sqrt{\bar{\alpha}_{\tau_{k-1}}}}$
11:         $z_{\tau_k}^{\mathrm{inv}} \leftarrow \sqrt{\bar{\alpha}_{\tau_k}}\, \hat{z}_{0,k-1}^{\mathrm{s}} + \sqrt{1 - \bar{\alpha}_{\tau_k}}\, \hat{\epsilon}_k^{\mathrm{s}}$
12:      **end for**
13:      Initialize $z_{\tau_K}^{\mathrm{t}} \leftarrow z_{\tau_K}^{\mathrm{inv}}$
14:      **for** $k = K, K-1, \ldots, 1$ **do**          ▷ Deterministic DDIM resynthesis under target branch
15:          $\hat{\epsilon}_k^{\mathrm{t}} \leftarrow \epsilon_{\theta_0 + \Delta\theta}\big(z_{\tau_k}^{\mathrm{t}}, \tau_k, S_i^{\mathrm{t}}, 1\big)$
16:         $\hat{z}_{0,k}^{\mathrm{t}} \leftarrow \dfrac{z_{\tau_k}^{\mathrm{t}} - \sqrt{1 - \bar{\alpha}_{\tau_k}}\, \hat{\epsilon}_k^{\mathrm{t}}}{\sqrt{\bar{\alpha}_{\tau_k}}}$
17:         $z_{\tau_{k-1}}^{\mathrm{t}} \leftarrow \sqrt{\bar{\alpha}_{\tau_{k-1}}}\, \hat{z}_{0,k}^{\mathrm{t}} + \sqrt{1 - \bar{\alpha}_{\tau_{k-1}}}\, \hat{\epsilon}_k^{\mathrm{t}}$
18:      **end for**
19:      Decode $\tilde{x}_i^{\mathrm{inv}\to\mathrm{t}} \leftarrow D_{\mathrm{VAE}}(z_{\tau_0}^{\mathrm{t}})$
20:      Add $(\tilde{x}_i^{\mathrm{inv}\to\mathrm{t}}, y_i^{\mathrm{s}})$ to $\widehat{\mathcal{D}}_{\mathrm{t}}^{\mathrm{inv}\to\mathrm{t}}$
21: **end for**
22: **return** $\widetilde{\mathcal{D}}_{\mathrm{t}}^{\mathrm{inv}\to\mathrm{t}}$

---

Table 5: Standard deviations (%) over three runs on Office-Home under the UDA setting.

| Method | Ar→Cl | Ar→Pr | Ar→Rw | Cl→Ar | Cl→Pr | Cl→Rw | Pr→Ar | Pr→Cl | Pr→Rw | Rw→Ar | Rw→Cl | Rw→Pr |
|---|---|---|---|---|---|---|---|---|---|---|---|---|
| ERM Vapnik (2013) | 0.25 | 0.26 | 0.42 | 0.17 | 0.20 | 0.15 | 0.07 | 0.17 | 0.05 | 0.34 | 0.33 | 0.01 |
| DANN Ganin et al. (2016) | 0.44 | 0.72 | 0.38 | 0.02 | 0.30 | 0.39 | 0.58 | 0.47 | 0.59 | 0.84 | 0.14 | 0.51 |
| CDAN Long et al. (2018) | 0.25 | 0.62 | 0.22 | 0.37 | 0.58 | 0.30 | 0.57 | 0.36 | 0.16 | 0.33 | 0.23 | 0.35 |
| AFN Xu et al. (2019) | 0.16 | 0.30 | 0.06 | 0.23 | 0.31 | 0.14 | 0.32 | 0.15 | 0.02 | 0.19 | 0.18 | 0.22 |
| MDD Zhang et al. (2019) | 0.51 | 0.32 | 0.06 | 0.24 | 0.73 | 0.41 | 0.36 | 0.53 | 0.24 | 0.03 | 0.09 | 0.11 |
| SDAT Rangwani et al. (2022) | 0.51 | 0.44 | 0.24 | 0.13 | 0.41 | 0.01 | 1.46 | 0.40 | 0.11 | 0.46 | 0.19 | 0.29 |
| MCC Jin et al. (2020) | 0.59 | 0.22 | 0.16 | 0.27 | 0.52 | 0.16 | 0.16 | 0.38 | 0.25 | 0.35 | 0.35 | 0.23 |
| ELS Zhang et al. (2023b) | 0.83 | 0.45 | 0.38 | 0.08 | 0.46 | 0.19 | 0.39 | 0.39 | 0.08 | 0.02 | 0.44 | 0.05 |
| **Diffusion-based methods** | | | | | | | | | | | | |
| MCC+Terra Zhuang et al. (2024) | 0.21 | 0.11 | 0.14 | 0.25 | 0.28 | 0.18 | 0.18 | 0.29 | 0.25 | 0.15 | 0.06 | 0.11 |
| ELS+Terra Zhuang et al. (2024) | 0.06 | 0.30 | 0.14 | 0.30 | 0.37 | 0.21 | 0.10 | 0.18 | 0.13 | 0.68 | 0.24 | 0.16 |
| MCC+VT-DUDA | 0.32 | 0.17 | 0.05 | 0.41 | 0.13 | 0.26 | 0.37 | 0.19 | 0.08 | 0.31 | 0.22 | 0.29 |
| ELS+VT-DUDA | 0.12 | 0.45 | 0.29 | 0.08 | 0.52 | 0.31 | 0.19 | 0.41 | 0.25 | 0.88 | 0.03 | 0.34 |

Table 6: Standard deviations (%) over three runs on Office-31 under the UDA setting.

| Method | A→W | D→W | W→D | A→D | D→A | W→A |
|---|---|---|---|---|---|---|
| ERM Vapnik (2013) | 0.11 | 0.00 | 0.00 | 1.22 | 0.15 | 0.11 |
| DANN Ganin et al. (2016) | 1.34 | 0.06 | 0.08 | 0.68 | 0.65 | 0.39 |
| CDAN Long et al. (2018) | 1.75 | 0.18 | 0.00 | 1.19 | 0.79 | 0.45 |
| AFN Xu et al. (2019) | 0.63 | 0.07 | 0.00 | 0.53 | 0.50 | 0.32 |
| MDD Zhang et al. (2019) | 1.00 | 0.15 | 0.00 | 0.10 | 0.68 | 0.18 |
| SDAT Rangwani et al. (2022) | 1.83 | 0.12 | 0.00 | 1.03 | 0.67 | 0.34 |
| MSGD Xia et al. (2022) | 0.50 | 0.30 | 0.00 | 0.30 | 0.40 | 0.50 |
| MCC Jin et al. (2020) | 0.38 | 0.08 | 0.09 | 1.47 | 0.50 | 0.20 |
| ELS Zhang et al. (2023b) | 0.51 | 0.06 | 0.00 | 0.20 | 0.54 | 0.16 |
| **Diffusion-based methods** | | | | | | |
| MCC+Terra Zhuang et al. (2024) | 0.06 | 0.06 | 0.00 | 0.09 | 0.18 | 0.12 |
| ELS+Terra Zhuang et al. (2024) | 0.17 | 0.06 | 0.00 | 0.48 | 0.28 | 0.11 |
| MCC+VT-DUDA | 0.12 | 0.10 | 0.00 | 0.04 | 0.18 | 0.15 |
| ELS+VT-DUDA | 0.15 | 0.12 | 0.00 | 0.10 | 0.20 | 0.18 |

### A.3 Standard Deviations of Main Experimental Results

To complement the mean accuracies reported in the main paper, we provide the standard deviations over three independent runs for the main UDA experiments on Office-Home, Office-31, and VisDA-2017. These results quantify the run-to-run stability of the compared methods under the same experimental protocol as in Sec. 4.

For transparency, we report standard deviations only for methods for which three-run records were available in our experimental logs. In particular, DCDM Zhang et al. (2025) is not included in this subsection because only the published mean accuracies were available to us.

### A.4 Additional Ablation Studies

To expand the pure-noise results summarized in Table 3 of the main paper, we report detailed transfer-wise results on Office-Home and Office-31, together with per-class results on VisDA-2017. Throughout this subsection, the reported **VT-DUDA** results are obtained using *only* pure-noise target-style generation under visual-token conditioning, without DDIM inversion. Concretely, VT-DUDA uses 50 generated images per class on Office-Home and Office-31, and 1000 generated images per class on VisDA-2017.

Table 7: Standard deviations (%) over three runs on VisDA-2017 under the UDA setting.

| Method | aero | bicycle | bus | car | horse | knife | motor | person | plant | skate | train | truck |
|---|---|---|---|---|---|---|---|---|---|---|---|---|
| ERM Vapnik (2013) | 9.90 | 2.64 | 3.26 | 2.20 | 1.35 | 3.60 | 1.41 | 1.03 | 1.80 | 3.97 | 0.79 | 0.67 |
| DANN Ganin et al. (2016) | 0.39 | 1.94 | 0.38 | 2.80 | 0.80 | 3.40 | 0.76 | 0.86 | 0.72 | 2.00 | 0.32 | 2.72 |
| CDAN Long et al. (2018) | 0.38 | 3.72 | 2.55 | 1.36 | 0.53 | 0.52 | 0.14 | 2.58 | 0.67 | 0.49 | 2.61 | 2.43 |
| AFN Xu et al. (2019) | 0.69 | 3.84 | 1.80 | 2.55 | 1.48 | 2.51 | 0.48 | 2.08 | 2.47 | 3.93 | 1.11 | 1.27 |
| MDD Zhang et al. (2019) | 2.40 | 9.46 | 1.18 | 0.66 | 0.85 | 4.01 | 0.65 | 1.81 | 1.21 | 4.30 | 1.58 | 0.33 |
| SDAT Rangwani et al. (2022) | 1.40 | 2.64 | 1.60 | 1.67 | 0.48 | 0.92 | 0.82 | 0.24 | 0.78 | 0.84 | 1.36 | 0.60 |
| MCC Jin et al. (2020) | 0.12 | 0.92 | 2.91 | 0.39 | 0.28 | 0.54 | 0.80 | 0.87 | 0.15 | 0.77 | 0.55 | 2.25 |
| ELS Zhang et al. (2023b) | 0.93 | 1.20 | 1.39 | 0.47 | 0.15 | 0.95 | 1.38 | 0.73 | 1.59 | 1.02 | 1.35 | 0.27 |
| **Diffusion-based methods** | | | | | | | | | | | | |
| MCC+Terra Zhuang et al. (2024) | 0.21 | 0.59 | 0.12 | 0.69 | 0.60 | 0.60 | 0.56 | 0.35 | 0.40 | 0.35 | 0.47 | 0.88 |
| ELS+Terra Zhuang et al. (2024) | 0.34 | 0.79 | 0.41 | 1.31 | 0.06 | 0.39 | 0.85 | 0.43 | 0.92 | 0.64 | 0.26 | 0.19 |
| MCC+VT-DUDA | 0.15 | 0.72 | 0.36 | 0.81 | 0.49 | 0.86 | 0.28 | 0.62 | 0.11 | 0.55 | 0.23 | 0.95 |
| ELS+VT-DUDA | 0.65 | 0.42 | 0.78 | 1.15 | 0.51 | 0.12 | 0.99 | 0.23 | 1.38 | 0.88 | 0.05 | 0.57 |

The reference diffusion baselines follow their respective comparison settings. On **Office-Home**, Terra is compared under the same pure-noise budget as VT-DUDA, namely 50 generated images per class, while DCDM is reported under its published setting with 200 generated images per class. On **Office-31**, the reported Terra result corresponds to its full configuration, which combines pure-noise generation with DDIM-inversion-based augmentation, whereas DCDM again uses 200 generated images per class. On **VisDA-2017**, the reported Terra result also corresponds to its full configuration with DDIM inversion, while DCDM uses 2000 generated images per class. We make these distinctions explicit because the purpose of this appendix is to assess the downstream effectiveness and sample efficiency of VT-DUDA under a lighter synthesis budget.

Tables 8 and 9 provide the detailed task-wise results corresponding to the averages reported in the main text. On Office-Home, the budget-matched comparison with Terra directly isolates the effect of conditioning quality: VT-DUDA improves over Terra on 11 out of 12 transfer tasks under both MCC and ELS. Compared with the larger-budget DCDM reference, ELS+VT-DUDA slightly improves over ELS+DCDM in average accuracy (73.72% vs. 73.64%), while MCC+VT-DUDA remains within 0.21 points of MCC+DCDM (73.50% vs. 73.71%), despite using only one quarter as many generated samples per class. This result supports the claim that the synthetic samples produced by VT-DUDA are more useful per sample for downstream adaptation.

On Office-31, the comparison is even stricter, since the Terra reference additionally benefits from DDIM-inversion-based augmentation and DCDM uses a larger synthetic-data budget. Even under this stronger comparison, VT-DUDA still achieves the best average accuracy under both downstream objectives: 91.37% for MCC and 91.92% for ELS, exceeding both Terra (91.34%, 91.28%) and DCDM (91.01%, 91.80%). Although the task-wise pattern is more mixed than on Office-Home, the average gain shows that the proposed visual-token conditioning improves the usefulness of the synthesized target-style data rather than merely increasing the number of generated samples.

Table 10 further reports the per-class results on VisDA-2017. Here VT-DUDA uses only pure-noise generation with 1000 images per class, while Terra uses its full configuration with DDIM inversion and DCDM uses 2000 images per class. Despite this lighter synthesis setting, VT-DUDA achieves the highest mean accuracy under both MCC and ELS. Relative to Terra, MCC+VT-DUDA improves 11 out of 12 categories, and ELS+VT-DUDA also improves 11 out of 12 categories. Relative to DCDM, MCC+VT-DUDA improves 7 out of 12 categories, while ELS+VT-DUDA improves 10 out of 12 categories. These class-wise results are consistent with the benchmark-level averages and indicate that the gains of VT-DUDA arise from systematically better synthetic supervision rather than from isolated categories.

## A.5 Additional Qualitative Results

In this subsection, we provide additional qualitative results to complement the quantitative findings reported in the main paper and in Appendix A.4. Our purpose is to visualize the behavior of the proposed visual-token-

Table 8: Detailed transfer accuracies (%) on Office-Home. VT-DUDA uses only pure-noise target-style generation with 50 generated images per class and no DDIM inversion. Terra is reported under the same pure-noise 50-per-class setting, while DCDM is included as a larger-budget reference with 200 generated images per class. Results are averaged over 12 transfer tasks (Avg). The best performance in each column is highlighted in **bold**.

| Method | Ar→Cl | Ar→Pr | Ar→Rw | Cl→Ar | Cl→Pr | Cl→Rw | Pr→Ar | Pr→Cl | Pr→Rw | Rw→Ar | Rw→Cl | Rw→Pr | Avg |
|---|---|---|---|---|---|---|---|---|---|---|---|---|---|
| ERM Vapnik (2013) | 44.06 | 67.12 | 74.26 | 53.26 | 61.96 | 64.54 | 51.91 | 38.90 | 72.94 | 64.51 | 43.84 | 75.39 | 59.39 |
| DANN Ganin et al. (2016) | 52.53 | 62.57 | 73.20 | 56.89 | 67.02 | 68.34 | 58.37 | 54.14 | 78.31 | 70.78 | 60.76 | 80.57 | 65.29 |
| CDAN Long et al. (2018) | 54.21 | 72.18 | 78.29 | 61.97 | 71.43 | 72.39 | 62.96 | 55.68 | 80.68 | 74.71 | 61.22 | 83.68 | 69.12 |
| AFN Xu et al. (2019) | 52.58 | 72.42 | 76.96 | 64.90 | 71.14 | 72.91 | 64.08 | 51.29 | 77.83 | 72.21 | 57.46 | 82.09 | 67.99 |
| MDD Zhang et al. (2019) | 56.37 | 75.53 | 79.17 | 62.95 | 73.21 | 73.55 | 62.56 | 54.86 | 79.49 | 73.84 | 61.45 | 84.06 | 69.75 |
| SDAT Rangwani et al. (2022) | 58.20 | 77.46 | 81.35 | 66.06 | 76.45 | 76.41 | 63.70 | 56.69 | 82.49 | 76.02 | 62.09 | 85.24 | 71.85 |
| MSGD Xia et al. (2022) | 58.70 | 76.90 | 78.90 | 70.10 | 76.20 | 76.60 | 69.00 | 57.20 | 82.30 | 74.90 | 62.70 | 84.50 | 72.40 |
| MCC Jin et al. (2020) | 56.83 | 79.81 | 82.66 | 67.80 | 77.02 | 77.82 | 66.98 | 55.43 | 81.79 | 73.95 | 61.41 | **85.44** | 72.24 |
| ELS Zhang et al. (2023b) | 57.79 | 77.65 | 81.62 | 66.59 | 76.74 | 76.43 | 62.69 | 56.69 | 82.12 | 75.63 | **62.85** | 85.35 | 71.84 |
| **Diffusion-based methods** | | | | | | | | | | | | | |
| MCC+Terra Zhuang et al. (2024) | 58.15 | 80.19 | 80.41 | 69.20 | 80.13 | 80.02 | 70.03 | 53.61 | 81.10 | 71.35 | 56.42 | 81.48 | 71.84 |
| ELS+Terra Zhuang et al. (2024) | 58.66 | 80.71 | 80.94 | 69.74 | 80.68 | 78.95 | 69.70 | 54.20 | 81.68 | 71.92 | 56.98 | 82.03 | 72.18 |
| MCC+DCDM Zhang et al. (2025) | 58.23 | 80.33 | **82.91** | 70.14 | 79.15 | 81.36 | 68.49 | 57.75 | **83.44** | 74.18 | 56.87 | **85.67** | 73.71 |
| ELS+DCDM Zhang et al. (2025) | **60.35** | 78.81 | 82.74 | 69.59 | 80.53 | 79.55 | 65.16 | **58.26** | 83.11 | **75.81** | 64.18 | 85.55 | 73.64 |
| MCC+VT-DUDA | 57.77 | **81.56** | 82.81 | 72.03 | **81.54** | 82.59 | **71.37** | 57.65 | 82.54 | 72.36 | 57.15 | 82.62 | 73.50 |
| ELS+VT-DUDA | 59.24 | 80.41 | 81.72 | **73.60** | 80.72 | **83.30** | 70.72 | 56.88 | 82.23 | 73.81 | 59.14 | 82.89 | **73.72** |

Table 9: Detailed transfer accuracies (%) on Office-31, with the corresponding VisDA-2017 mean accuracy (%) reported in the last column. VT-DUDA uses only pure-noise target-style generation with 50 generated images per class and no DDIM inversion. Terra is reported under its full configuration combining pure-noise generation (50 per class) with DDIM-inversion-based augmentation, while DCDM is included under its published 200-per-class setting. The best performance in each column is highlighted in **bold**.

| Dataset | Office-31 | | | | | | | VisDA-2017 |
|---|---|---|---|---|---|---|---|---|
| Method | A→W | D→W | W→D | A→D | D→A | W→A | Avg | mean |
| ERM Vapnik (2013) | 77.07 | 96.60 | 99.20 | 81.08 | 64.11 | 64.01 | 80.35 | 51.47 |
| DANN Ganin et al. (2016) | 89.85 | 97.95 | 99.90 | 83.26 | 73.28 | 73.75 | 86.33 | 79.02 |
| CDAN Long et al. (2018) | 92.42 | 98.62 | **100.00** | 91.44 | 74.61 | 72.80 | 88.32 | 80.74 |
| AFN Xu et al. (2019) | 91.82 | 98.77 | **100.00** | 95.12 | 72.43 | 70.71 | 88.14 | 74.64 |
| MDD Zhang et al. (2019) | 93.55 | 98.66 | **100.00** | 93.92 | 75.29 | 73.95 | 89.23 | 81.10 |
| SDAT Rangwani et al. (2022) | 91.32 | 98.83 | **100.00** | 95.25 | 76.97 | 73.19 | 89.26 | 83.23 |
| MSGD Xia et al. (2022) | 95.50 | 99.20 | **100.00** | 95.60 | 77.30 | 77.00 | 90.80 | 84.60 |
| MCC Jin et al. (2020) | 94.09 | 98.32 | 99.67 | 94.25 | 75.89 | 75.46 | 89.61 | 83.32 |
| ELS Zhang et al. (2023b) | 93.84 | 98.78 | **100.00** | 95.78 | 77.72 | 75.13 | 90.21 | 83.40 |
| **Diffusion-based methods** | | | | | | | | |
| MCC+Terra Zhuang et al. (2024) | 94.55 | 99.03 | **100.00** | 96.46 | 78.64 | 79.37 | 91.34 | 85.39 |
| ELS+Terra Zhuang et al. (2024) | 94.09 | **99.21** | **100.00** | 96.25 | 78.67 | 79.45 | 91.28 | 86.86 |
| MCC+DCDM Zhang et al. (2025) | 95.51 | 98.58 | 99.93 | 95.31 | 78.26 | 78.43 | 91.01 | 86.56 |
| ELS+DCDM Zhang et al. (2025) | 96.90 | 98.91 | **100.00** | 97.46 | **79.79** | 77.74 | 91.80 | 86.29 |
| MCC+VT-DUDA | 96.22 | 98.10 | 99.95 | **99.88** | 77.01 | 77.05 | 91.37 | 86.92 |
| ELS+VT-DUDA | **97.58** | 97.52 | 99.52 | 98.02 | 77.88 | **81.01** | **91.92** | **87.56** |

conditioned generation framework under the two synthetic-data construction routes used in VT-DUDA, namely pure-noise target-style synthesis and DDIM-inversion-based target-style translation. We further provide an additional visualization of inference-time token-strength control.

We begin with qualitative results obtained using Algorithm 1. In this setting, a labeled source image provides both the class prompt and the corresponding visual-token conditioning, and the target-style sample is synthesized from Gaussian noise under the target branch ($\sigma = 1$). Figure 5 summarizes representative pure-noise synthesis results on Office-Home. The figure contains all twelve transfer tasks, with one representative class shown for each domain shift. Figure 6 provides the corresponding qualitative visualization on VisDA-2017, where six representative examples are shown in a single composed figure.

Table 10: Per-class transfer accuracies (%) on VisDA-2017. VT-DUDA uses only pure-noise target-style generation with 1000 generated images per class and no DDIM inversion. Terra is reported under its full configuration with DDIM-inversion-based augmentation, while DCDM is included under its published 2000-per-class setting. The best performance in each column is highlighted in **bold**.

| Method | aero | bicycle | bus | car | horse | knife | motor | person | plant | skate | train | truck | mean |
|---|---|---|---|---|---|---|---|---|---|---|---|---|---|
| ERM Vapnik (2013) | 81.71 | 22.46 | 54.08 | **76.21** | 74.83 | 10.69 | 83.81 | 18.71 | 80.88 | 28.66 | 79.66 | 5.98 | 51.47 |
| DANN Ganin et al. (2016) | 94.75 | 73.47 | 83.46 | 47.91 | 87.00 | 88.30 | 88.47 | 77.18 | 88.16 | 90.05 | 87.21 | 42.26 | 79.02 |
| CDAN Long et al. (2018) | 94.55 | 74.41 | 82.22 | 58.92 | 90.56 | 96.22 | 89.71 | 78.90 | 86.11 | 89.06 | 84.81 | 43.42 | 80.74 |
| AFN Xu et al. (2019) | 93.13 | 54.76 | 81.03 | 69.74 | 92.36 | 75.88 | 92.11 | 73.83 | 93.16 | 55.55 | **90.48** | 23.63 | 74.64 |
| MDD Zhang et al. (2019) | 92.68 | 65.26 | 82.29 | 66.78 | 91.68 | 92.09 | **93.18** | 79.67 | 92.12 | 84.95 | 83.85 | 48.66 | 81.10 |
| SDAT Rangwani et al. (2022) | 94.51 | 83.56 | 74.28 | 65.78 | 93.00 | 95.83 | 89.61 | 80.04 | 90.86 | 91.47 | 84.95 | 54.93 | 83.23 |
| MSGD Xia et al. (2022) | 97.50 | 83.40 | **84.40** | 69.40 | 95.90 | 94.10 | 90.90 | 75.50 | **95.50** | 94.60 | 88.10 | 44.90 | 84.60 |
| MCC Jin et al. (2020) | 95.26 | 86.14 | 77.12 | 69.98 | 92.83 | 94.84 | 86.52 | 77.78 | 90.26 | 90.98 | 85.68 | 52.52 | 83.32 |
| ELS Zhang et al. (2023b) | 94.76 | 83.38 | 75.44 | 66.45 | 93.16 | 95.14 | 89.09 | 80.13 | 90.77 | 91.06 | 84.09 | 57.36 | 83.40 |
| **Diffusion-based methods** | | | | | | | | | | | | | |
| MCC+Terra Zhuang et al. (2024) | 96.20 | 87.27 | 78.77 | 70.59 | 94.18 | 95.49 | 85.08 | 85.48 | 92.24 | 93.20 | 86.26 | 59.88 | 85.39 |
| ELS+Terra Zhuang et al. (2024) | 95.98 | 87.12 | 81.60 | 70.84 | 95.14 | 96.29 | 88.47 | 87.78 | 94.75 | 94.06 | 86.47 | **63.83** | 86.86 |
| MCC+DCDM Zhang et al. (2025) | 96.43 | 87.28 | 83.16 | 74.37 | 94.59 | 96.13 | 88.60 | 81.90 | 92.98 | 94.45 | 87.26 | 61.56 | 86.56 |
| ELS+DCDM Zhang et al. (2025) | 96.20 | 84.79 | 83.15 | 73.28 | 94.76 | 96.58 | 90.99 | 82.21 | 92.98 | 93.37 | 87.49 | 59.70 | 86.29 |
| MCC+VT-DUDA | 97.11 | **88.49** | 82.36 | 72.34 | 95.88 | 96.42 | 89.23 | **88.07** | 94.91 | 94.16 | 87.14 | 56.93 | 86.92 |
| ELS+VT-DUDA | **97.79** | 87.91 | 83.19 | 73.14 | **96.06** | **96.81** | 91.25 | 88.01 | 95.15 | **94.71** | 88.94 | 57.76 | **87.56** |

We then present qualitative results obtained using Algorithm 2. In this setting, a labeled source image is first inverted under the source branch ($\sigma = 0$), and is then translated to the target style under the target branch ($\sigma = 1$) using the same class prompt together with the visual tokens extracted from the source image. Figure 7 shows representative results on Office-Home for all twelve transfer tasks, again using one representative class per transfer direction.

Finally, Figure 8 visualizes inference-time token-strength scaling for synthesis in the Art→Clipart setting. Here, we extract visual tokens from a class-matched source image and generate target-style samples under the target condition ($\sigma = 1$), while varying the token strength $\eta_0 \in \{0.5, 1.0, 1.5, 2.0\}$. This figure illustrates that the proposed tokenized conditioning interface permits direct control of the conditioning signal at inference time without modifying the diffusion backbone or retraining the model.

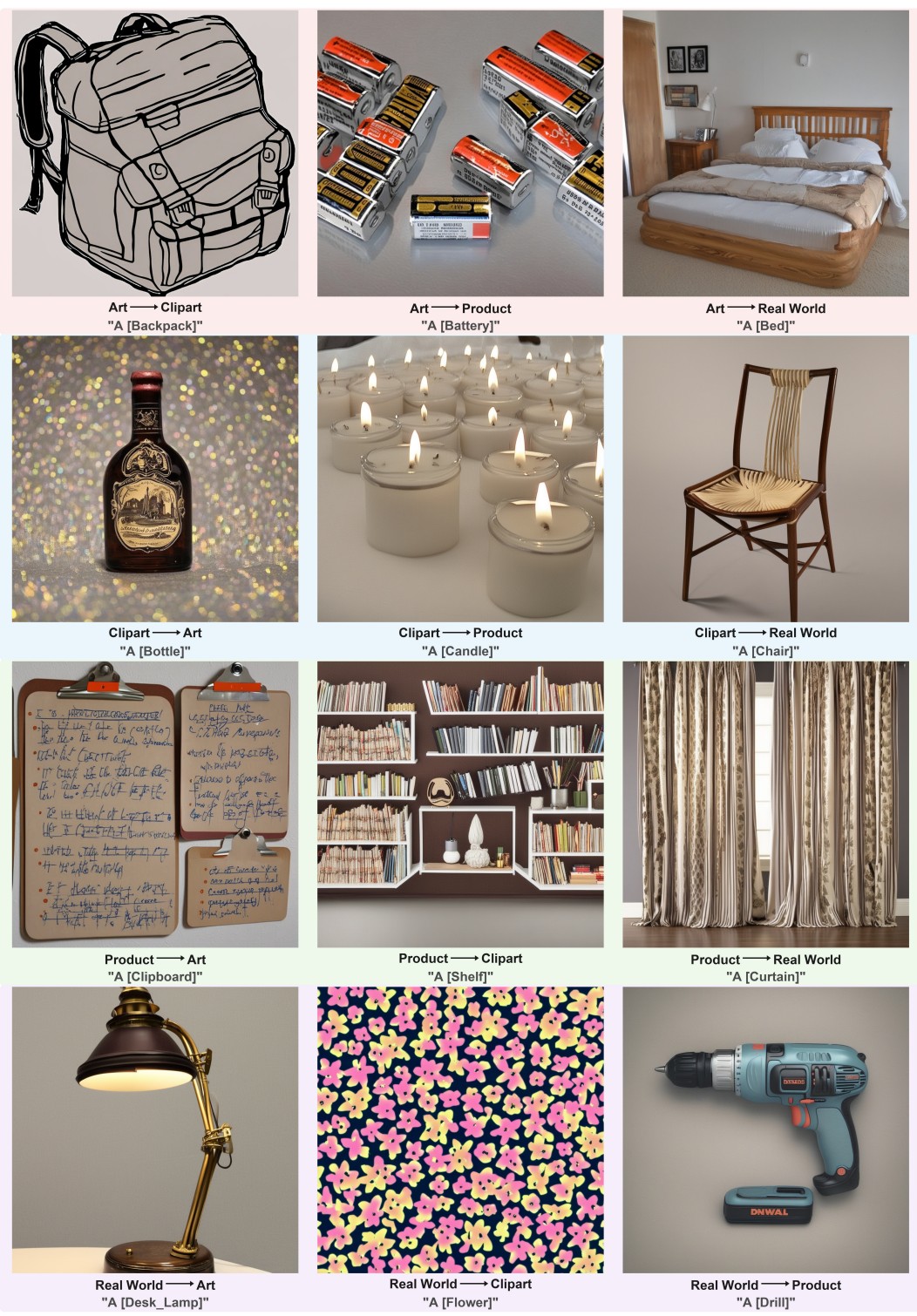

Figure 5: Additional qualitative results for pure-noise target-style synthesis on Office-Home using Algorithm 1.

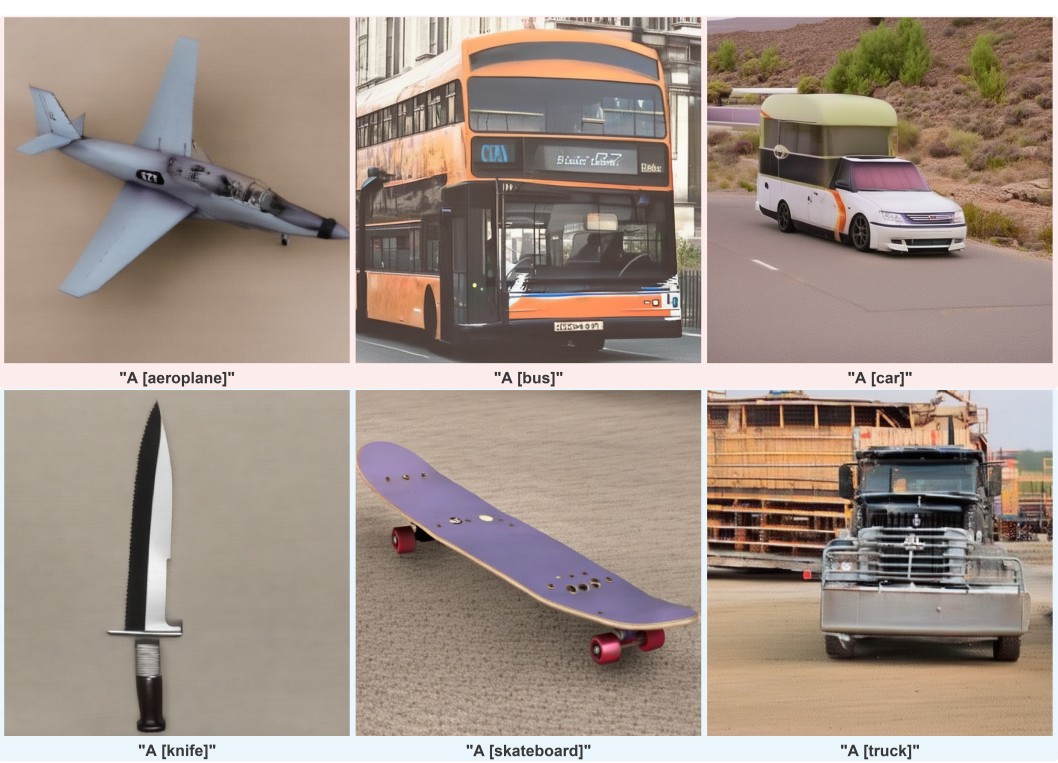

Figure 6: Additional qualitative results for pure-noise target-style synthesis on VisDA-2017 using Algorithm 1.

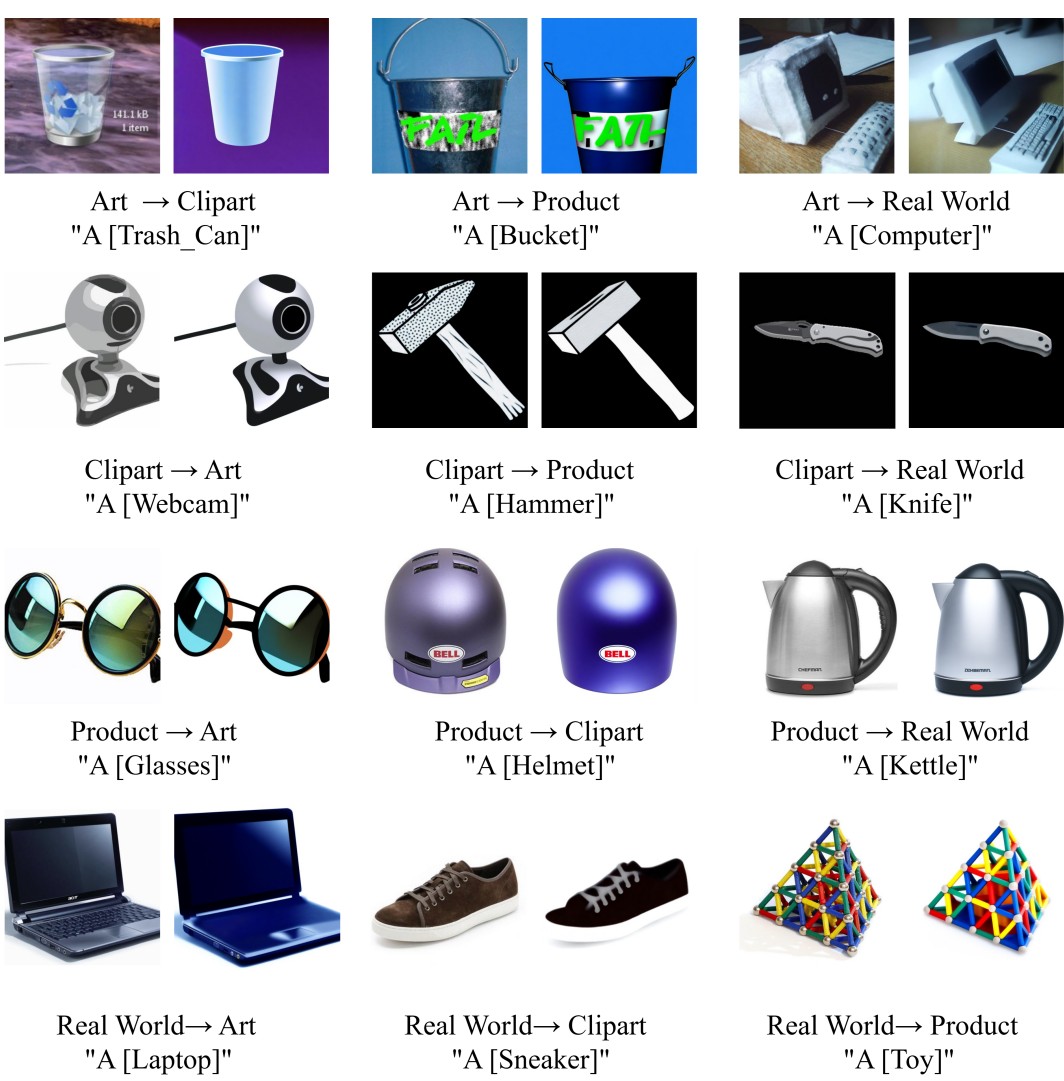

Figure 7: Additional qualitative results for DDIM-inversion-based target-style translation on Office-Home using Algorithm 2.

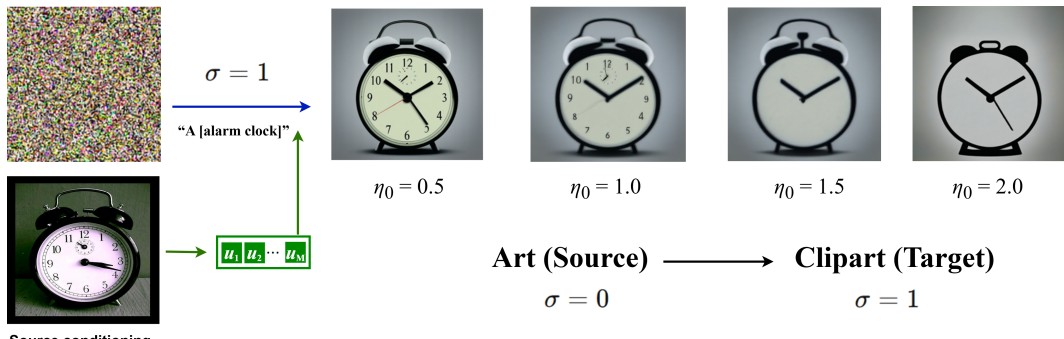

Figure 8: Token-strength scaling for synthesis (Art→Clipart). We extract visual tokens from a class-matched source image and generate target-style samples under the target condition $\sigma=1$, while varying the token strength $\eta_0 \in \{0.5, 1.0, 1.5, 2.0\}$.

## A.6 Comparison with the SDXL Prior

To further clarify the role of diffusion adaptation in generation-based UDA, we include an additional comparison against target-like sample generation that relies only on the prior knowledge of a pretrained SDXL model. The goal of this comparison is to distinguish improvements obtained from *stronger prompt design and sample selection on top of an off-the-shelf text-to-image prior* from improvements obtained by *explicitly adapting the generative model and its conditioning interface for the source-target transfer*. To this end, we adopt the SDXL-prior baselines reported in Terra and compare them with our method under the same Office-Home benchmark and the same ELS downstream UDA setting.

For clarity, we briefly summarize how each SDXL-prior baseline is constructed. These baselines synthesize target-like samples directly from the SDXL prior with progressively stronger prompt engineering and post-selection:

- **SDXL (random):** Samples are generated using only the class prompt "A [CLASS]", where [$CLASS$] denotes the category label.

- **SDXL (styles):** First, GPT-4 is asked to generate 50 diverse style prompts. These style prompts are then inserted into the template "A [CLASS], an everyday object in office and home, in the style of [STYLE]" to synthesize images.

- **SDXL (target):** Based on the previous setting, the style placeholder [$STYLE$] is replaced by the target-domain name itself, e.g., "Clipart", so that the SDXL prior is prompted more directly toward the target domain.

- **SDXL (target styles):** Instead of using only the domain name, GPT-4 is asked to generate 50 prompts that explicitly describe the target-domain style, yielding richer target-style prompts for synthesis.

- **SDXL (selected):** Finally, the samples generated by *SDXL (target styles)* are further filtered with a confidence-based active-learning-style selection strategy to remove poor-quality or misclassified synthetic samples.

This sequence of baselines forms a progressively stronger test of what can be achieved by the SDXL prior alone. It starts from plain class prompting, then adds generic style diversity, explicitly injects target-domain information, refines that information into more detailed target-style prompts, and finally applies sample selection to improve the quality of the synthetic set.

The results are shown in Table 11. As the SDXL-prior baselines become stronger, the average UDA accuracy increases from 69.92% for *SDXL (random)* to 73.99% for *SDXL (selected)*. Our method achieves 76.74% average accuracy, which is +2.75 points higher than the strongest SDXL-prior baseline. In addition, ELS+VT-DUDA obtains the best result on 10 out of 12 transfer tasks.

These results show that the proposed visual-token-conditioned adaptation improves the downstream usefulness of synthetic data beyond prompt engineering and sample selection on top of the SDXL prior. The comparison with the SDXL prior supports our claim that, in diffusion-guided UDA, the utility of synthetic data depends not only on the strength of the pretrained generative prior, but also on the design of the adaptation and conditioning mechanism.

## A.7 Additional Diagnostic Experiments

We conduct additional diagnostic experiments to address the questions about the role of visual-token conditioning. These experiments are performed under the inversion-free Office-Home setting using ELS Zhang et al. (2023b) as the downstream UDA objective. For the quantitative diagnostics, we use two representative transfer tasks: Ar→Cl and Cl→Rw. Ar→Cl is a challenging transfer task that is also used in our generation-budget ablation, while Cl→Rw is used in our token-count ablation and provides a complementary target domain.

Table 11: Comparison with SDXL-prior-based target-like sample generation on Office-Home under the UDA setting with ELS. The five SDXL-prior baselines are taken from Terra Zhuang et al. (2024), while the last row reports our method under the same benchmark and downstream UDA setting. The best result in each column is highlighted in **bold**.

| Method | Ar→Cl | Ar→Pr | Ar→Rw | Cl→Ar | Cl→Pr | Cl→Rw | Pr→Ar | Pr→Cl | Pr→Rw | Rw→Ar | Rw→Cl | Rw→Pr | Avg |
|---|---|---|---|---|---|---|---|---|---|---|---|---|---|
| SDXL (random) | 56.88 | 73.64 | 80.38 | 69.18 | 73.64 | 80.40 | 68.93 | 56.54 | 80.38 | 68.93 | 56.54 | 73.64 | 69.92 |
| SDXL (styles) | 55.23 | 77.09 | 80.26 | 68.11 | 77.09 | 80.26 | 68.11 | 55.23 | 80.45 | 68.11 | 55.23 | 77.04 | 70.18 |
| SDXL (target) | 59.70 | 75.51 | 82.26 | 66.67 | 75.51 | **82.26** | 66.67 | 59.70 | 82.26 | 66.67 | 59.70 | 75.51 | 71.04 |
| SDXL (target styles) | 60.76 | 79.52 | 81.68 | 70.95 | 79.52 | 81.68 | 70.95 | 60.76 | 81.68 | 70.95 | 60.76 | 79.52 | 73.23 |
| SDXL (selected) | 61.63 | 79.81 | 82.19 | 71.98 | 79.73 | 81.82 | **71.69** | 61.58 | 82.07 | 72.76 | 62.15 | 80.42 | 73.99 |
| ELS+VT-DUDA | **63.26** | **83.74** | **85.03** | **72.63** | **85.67** | 78.89 | 71.55 | **64.91** | **85.29** | **77.58** | **65.17** | **87.16** | **76.74** |

Table 12: Controlled conditioning ablation on Office-Home under the inversion-free protocol with ELS as the downstream UDA objective. All generated-data variants use the same backbone, sampler, generation budget, and ELS-based downstream training protocol. "Target-side tokens" indicates whether visual tokens are used when training the target branch of the diffusion model. The original ELS baseline is included as a discriminative UDA reference and does not use diffusion-generated synthetic data.

| Method | Visual conditioning | Target-side tokens | Ar→Cl | Cl→Rw |
|---|---|---|---|---|
| ELS Zhang et al. (2023b) | — | — | 57.79 | 76.43 |
| ELS+Text-only baseline | none | no | 57.12 | 78.23 |
| ELS+Single-global-token | one projected global feature | yes | 58.68 | 80.98 |
| ELS+Repeated-global-token | same global feature repeated 10 times | yes | 57.26 | 79.31 |
| ELS+VT-DUDA w/o target-side tokens | 10 token-specific visual tokens | no | 56.82 | 78.69 |
| ELS+VT-DUDA (Ours) | 10 token-specific visual tokens | yes | **59.24** | **83.30** |

Unless otherwise stated, all generated-data variants use the same SDXL backbone, LoRA adaptation strategy, image resolution, sampler, number of inference steps, generation budget, and ELS-based downstream classifier-training protocol. These diagnostics are intended to examine: (i) whether the proposed visual-token interface is more effective than simpler global image-conditioning baselines, (ii) whether target-side visual tokens contribute to the token-conditioned training interface, (iii) how often generated samples are classified as their inherited source labels by an independent evaluator, and (iv) how visual tokens are reflected in cross-attention responses during generation.

### A.7.1 Controlled Comparison with Simpler Image-conditioning Baselines

To test whether the improvement can be explained simply by adding an image feature to the conditioning sequence, we compare ELS+VT-DUDA with two simpler image-conditioning baselines. In the *single-global-token* baseline, the source image is encoded into one global visual feature, which is linearly projected to the cross-attention dimension and concatenated with the class-prompt tokens as a single additional conditioning token. In the *repeated-global-token* baseline, the same projected global feature is repeated across $M = 10$ visual-token positions and concatenated with the class-prompt tokens. That is, its visual context is $[g; g; \ldots; g]$, where the same global feature $g$ is copied into all 10 token slots. This baseline controls for the number of visual-token positions while removing token-specific variation. All generated-data variants use the same generation budget and downstream ELS protocol.

Table 12 shows several trends. First, the text-only diffusion baseline is not uniformly better than the original ELS baseline: it is slightly lower on Ar→Cl but higher on Cl→Rw. This indicates that diffusion-generated data without visual-token conditioning is not sufficient to explain the gains of VT-DUDA. Second, adding a single global image token improves over the text-only baseline on both diagnostic tasks, suggesting that image-dependent conditioning is useful in this setting. Third, repeating the same global feature across 10 token positions does not match the performance of VT-DUDA, even though it uses the same number of visual-token slots. This suggests that the improvement of VT-DUDA is not explained only by increasing the length of the conditioning sequence. Finally, removing target-side visual tokens from VT-DUDA reduces performance on both tasks relative to the full model. This supports the design motivation that target-side visual tokens help train the target branch under a token-augmented conditioning interface. We interpret

Table 13: Post-hoc label-consistency analysis of generated Office-Home samples. For each transfer, an independent ResNet-50 oracle is trained on real images from the corresponding target domain and used only for evaluation. The downstream ELS+VT-DUDA accuracy is reported as a reference for the final UDA outcome under the inversion-free protocol.

| Generated set | Target-domain evaluator | LC (%) | ELS+VT-DUDA Acc. (%) |
|---|---|---|---|
| Ar→Cl | Clipart oracle | 54.12 | 59.24 |
| Cl→Rw | Real-World oracle | 82.57 | 83.30 |

these results as evidence that the proposed token-specific conditioning is useful for these representative Office-Home transfers.

### A.7.2 Post-hoc Label-consistency Analysis

We next evaluate whether generated Office-Home samples preserve their intended class semantics under inherited source labels. This analysis is independent of the generative training procedure: the evaluator is not used for diffusion training, LoRA adaptation, image-to-token encoder learning, image generation, or downstream ELS training. It is used only after generation to measure label consistency.

For each target domain, we train an independent target-domain oracle classifier using real Office-Home images from that domain and their ground-truth class labels. Specifically, for Ar→Cl, we train a Clipart-domain evaluator on real Clipart images. For Cl→Rw, we train a Real-World-domain evaluator on real Real-World images. Each evaluator is a ResNet-50 classifier trained for the 65 Office-Home categories. The generated samples are then evaluated using the oracle corresponding to their target domain: Ar→Cl samples are evaluated by the Clipart oracle, and Cl→Rw samples are evaluated by the Real-World oracle. Target labels are used only for this post-hoc diagnostic and are not used by VT-DUDA or any baseline during UDA training.

Let $\widetilde{\mathcal{D}}_{s \to t}$ denote the generated image set for source domain $s$ and target domain $t$. Each generated image $\tilde{x}$ inherits its class label $\tilde{y}$ from the source conditioning image. Given a target-domain evaluator $h_t$, label consistency is defined as

$$\mathrm{LC}(s \to t) = \frac{1}{|\widetilde{\mathcal{D}}_{s \to t}|} \sum_{(\tilde{x}, \tilde{y}) \in \widetilde{\mathcal{D}}_{s \to t}} \mathbf{1}\left[h_t(\tilde{x}) = \tilde{y}\right] \times 100\%. \tag{36}$$

Here, $h_t$ is trained only on real images from the target domain $t$ and is used only to evaluate generated samples after generation.

Table 13 reports the resulting label-consistency scores for the generated VT-DUDA samples, together with the corresponding downstream ELS+VT-DUDA target accuracies for context. These two quantities measure different aspects of the pipeline. LC measures the fraction of generated samples whose inherited label is predicted by an independently trained target-domain evaluator, while the downstream accuracy measures the final target-domain performance of a classifier trained with the synthetic data and the ELS objective.

The two transfer tasks exhibit different consistency levels: Cl→Rw obtains a relatively high LC score of 82.57%, whereas Ar→Cl obtains a more moderate score of 54.12%. The same task-dependent pattern is also reflected in the downstream accuracies, with Cl→Rw achieving 83.30% and Ar→Cl achieving 59.24%. However, we do not interpret the downstream accuracy as an upper bound or direct substitute for LC, since the downstream classifier is trained with an ELS-based UDA objective and is evaluated on real target-domain data, whereas LC is a post-hoc oracle-based diagnostic on generated samples.

Overall, this analysis suggests that the generated data can preserve class information to different degrees depending on the target domain and transfer direction. The evaluator predictions are never used as supervision or feedback for the generative model or downstream UDA training, and LC should be interpreted only as a post-hoc semantic diagnostic rather than as a definitive measure of label correctness.

Table 14: Sensitivity to the image-to-token encoder architecture on Office-Home under the inversion-free protocol with ELS as the downstream UDA objective. All VT-DUDA variants use the same token count, diffusion backbone, sampler, generation budget, and downstream training protocol; only the image-to-token encoder backbone is changed.

| Method | Ar→Cl | Cl→Rw |
|---|---|---|
| ELS Zhang et al. (2023b) | 57.79 | 76.43 |
| ELS+VT-DUDA (ResNet-18 encoder) | **59.24** | 83.30 |
| ELS+VT-DUDA (ViT-Tiny encoder) | 58.71 | **83.68** |

### A.7.3 Same-class Source-instance Diversity

To illustrate instance-level conditioning qualitatively, we generate target-style samples from different source images belonging to the same class while keeping the class prompt, target branch, sampler, and generation protocol fixed. In this visualization, the source image used to extract visual tokens is the only intended changing factor across rows. This setup examines whether different source instances within the same class can induce different generated samples under the same class-level prompt.

Figure 9 shows that different source instances from the same class can lead to visibly different generated samples under the same class prompt. This observation is consistent with the motivation of VT-DUDA: visual tokens make the conditioning depend on the source instance rather than only on the class prompt. We do not interpret this visualization as evidence that individual visual tokens are semantically disentangled or directly interpretable.

### A.7.4 Cross-attention Heatmap Visualization of Visual Tokens

To further inspect how visual tokens participate in generation, we visualize the cross-attention responses associated with the 10 visual tokens. For each visual token $u_i$, we collect the cross-attention probability assigned to that token from the recorded cross-attention layers and denoising steps during target generation. We average these responses over the recorded heads, layers, and steps, upsample the resulting spatial map to image resolution, and overlay it on the generated image.

This visualization is intended as a token-response diagnostic. It should not be interpreted as proving that each individual token corresponds to a stable human-nameable semantic part. Instead, it shows whether different visual tokens are associated with different spatial attention patterns during generation.

Figure 10 shows that the recorded visual-token attention responses are spatially non-uniform and vary across tokens. This suggests that the visual-token sequence is not ignored by the cross-attention mechanism in this example. However, we interpret the heatmaps conservatively: they provide qualitative evidence of token-dependent spatial response patterns, but they do not establish a one-to-one correspondence between individual tokens and semantic object parts.

### A.7.5 Sensitivity to the Image-to-token Encoder Architecture

We further examine whether VT-DUDA is sensitive to the architecture of the image-to-token encoder $G_\psi$. The main experiments use a ResNet-18 encoder for this component. To test whether replacing the convolutional encoder with a transformer-based encoder changes the downstream behavior, we replace the ResNet-18 encoder with a ViT-Tiny encoder, i.e., the smallest ViT variant used in our implementation, while keeping the rest of the pipeline unchanged. Specifically, the diffusion backbone, LoRA adaptation strategy, token count, sampler, generation budget, inversion-free protocol, and ELS-based downstream UDA training are kept the same. We evaluate the resulting variant on the same two Office-Home transfer tasks used in the diagnostic experiments, Ar→Cl and Cl→Rw.

Table 14 shows that replacing the ResNet-18 image-to-token encoder with a ViT-Tiny encoder leads to similar downstream UDA performance on the two diagnostic tasks. The ViT-Tiny variant is slightly lower on Ar→Cl and slightly higher on Cl→Rw. Both VT-DUDA variants improve over the ELS baseline on these tasks. These results suggest that, in this setting, the downstream performance is not strongly affected by

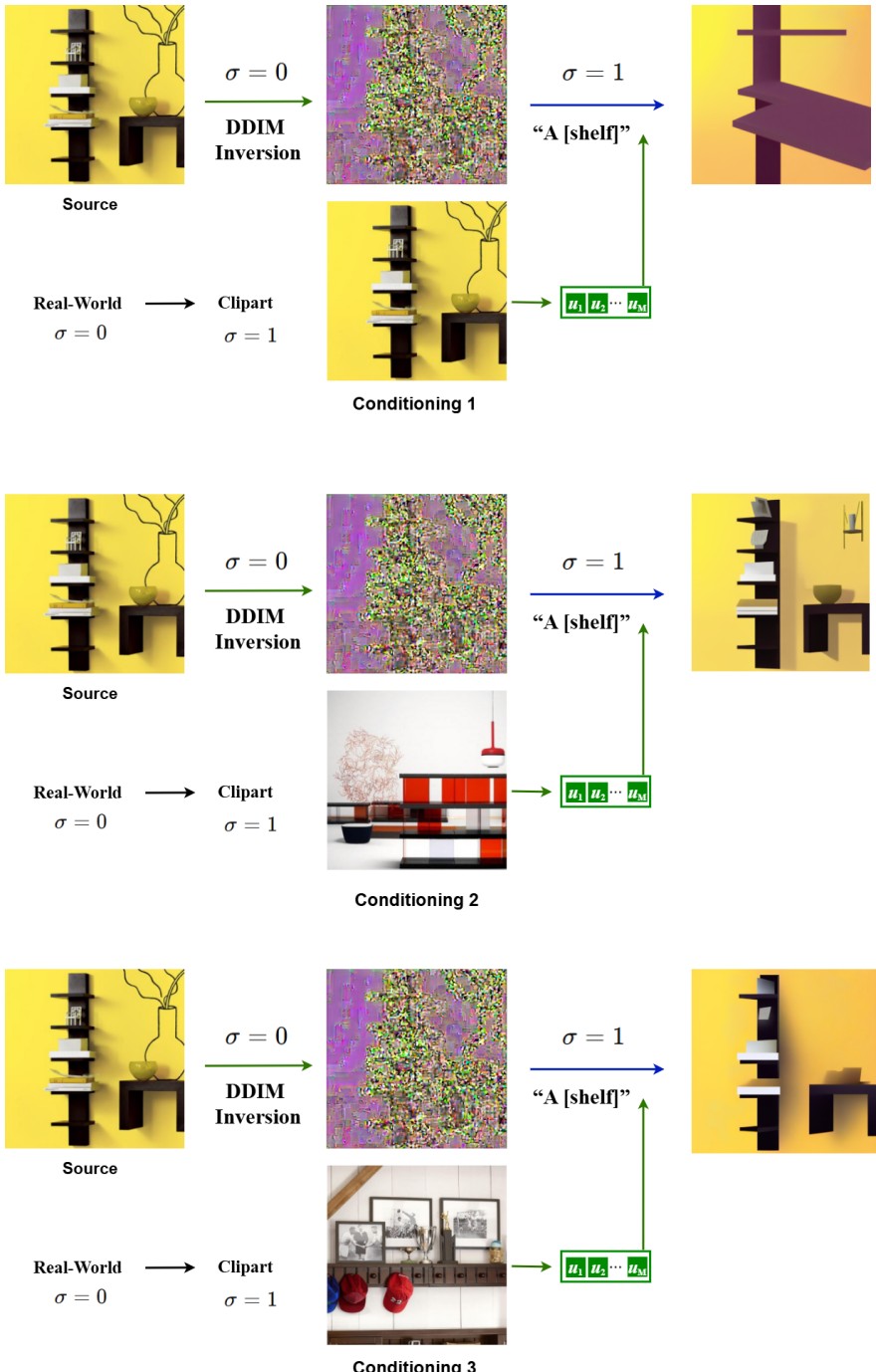

Figure 9: Same-class source-instance diversity on Office-Home Rw→Cl. Each row uses the same class prompt and the same target branch, while changing only the source image used to extract visual tokens. The generated target-style samples show visible within-class variation. This visualization is intended to illustrate instance-dependent conditioning behavior, not to claim that individual visual tokens correspond to human-interpretable semantic parts.

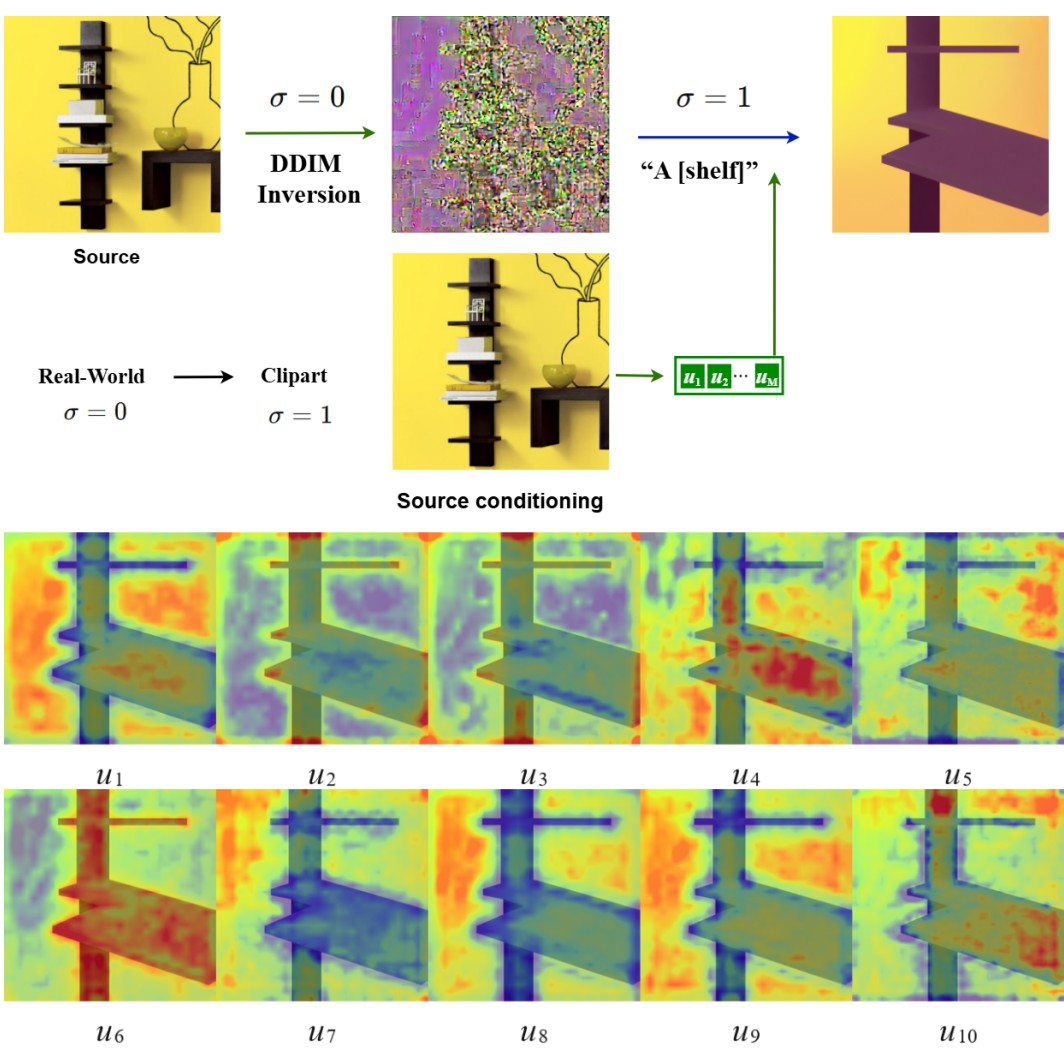

Figure 10: Cross-attention heatmap visualization for the 10 visual tokens under DDIM inversion-based target generation. The source image is an Office-Home *Real World* shelf image, and generation is conditioned on the text prompt "a shelf" together with ten image-derived visual tokens $\{u_i\}_{i=1}^{10}$. Each panel corresponds to one visual token $u_i$ and shows its aggregated cross-attention map overlaid on the generated image. Warmer colors indicate spatial regions that assign higher attention probability to the corresponding visual token, while cooler colors indicate lower attention. The maps are averaged over the recorded cross-attention layers and denoising steps during target generation. This visualization is a token-response diagnostic and does not assume that individual tokens are semantically disentangled or directly interpretable.

Table 15: Diffusion-adaptation training cost. Both methods are trained for 5000 optimization steps on the same GPU class. Wall-clock time is measured under our implementation and hardware setting.

| Method | GPU | Training steps | Training time |
|---|---|---|---|
| Terra UDA | A100 80GB | 5000 | 98.18 h |
| VT-DUDA | A100 80GB | 5000 | 112.25 h |

Table 16: Runtime and GPU-memory profiling comparison. Peak allocated and peak reserved memory are measured using CUDA memory counters. Single-image generation latency is reported in seconds per image.

| Metric | Terra UDA | VT-DUDA |
|---|---|---|
| Peak GPU memory for training | 33,105.29 MiB / 32.33 GiB | 48,177.98 MiB / 47.05 GiB |
| Peak reserved GPU memory for training | 40,616.00 MiB / 39.66 GiB | 54,442.00 MiB / 53.17 GiB |
| Pure-noise generation time per image | 1.75 s | 3.08 s |
| Pure-noise peak GPU memory | 9,178.33 MiB / 8.96 GiB | 20,372.59 MiB / 19.90 GiB |
| DDIM source-to-target total time per image | 3.42 s | 4.54 s |
| DDIM peak GPU memory | 9,175.36 MiB / 8.96 GiB | 22,075.05 MiB / 21.56 GiB |

replacing the ResNet-18 encoder with the smallest ViT variant used in our implementation. We therefore keep ResNet-18 as the default encoder in the main experiments because it provides competitive performance with a lightweight design.

## A.8 Computational Cost

We report the computational cost of VT-DUDA and Terra UDA in Table 15 and Table 16. All measurements were collected on NVIDIA A100 80GB GPUs using our implementations. Table 15 summarizes diffusion-adaptation training time under the same 5000-step budget, while Table 16 reports diagnostic measurements of GPU memory usage and single-image generation latency.

Table 15 shows that VT-DUDA requires a longer diffusion-adaptation training time than Terra UDA under the same 5000-step budget. Table 16 further shows that VT-DUDA uses more GPU memory during both training and generation, and has higher measured single-image generation latency in both pure-noise generation and DDIM source-to-target generation. These overheads are expected because VT-DUDA introduces an image-to-token encoder and token-augmented conditioning while keeping the pretrained diffusion backbone and denoising objective unchanged. We include these measurements to make the practical cost of the additional conditioning interface explicit.

