# OpenReview forum: "VT-DUDA: Visual Token Conditioning for Diffusion-guided Unsupervised Domain Adaptation"
_TMLR — Accepted by TMLR_

### Review · Reviewer_GmZJ · 2026-03-30

**Summary Of Contributions:**

This paper proposes VT-DUDA, a visual-token conditioning framework for diffusion-guided unsupervised domain adaptation (UDA). The key idea is to augment the standard text-based conditioning in latent diffusion models with instance-level visual tokens extracted from source and target images. These tokens are concatenated with text embeddings and injected into the cross-attention layers, enabling more fine-grained conditioning during both training and generation. The method is applied to generate labeled target-style data, which is then used to train a downstream classifier.

**Additional Comments:**

I want to mention again that I am not a CV researcher but focus more on the DA perspective.

**Audience:**

Yes

**Audience Explanation:**

Those CV people in the domain adaptation field will absolutely be.interested.

**Broader Impact Concerns:**

NO concerns.

**Claims And Evidence:**

Yes

**Claims Explanation:**

- Well-motivated problem formulation

The paper identifies a meaningful limitation in existing diffusion-based UDA methods, namely the lack of instance-level conditioning when relying solely on class prompts. This is a valid and relevant problem.

- Simple yet effective design

The proposed visual-token conditioning mechanism is conceptually straightforward and integrates cleanly into existing diffusion frameworks without modifying the backbone or objective.

- Strong empirical evaluation

The experiments are comprehensive:
[1] Multiple standard UDA benchmarks are used.
[2] Comparisons include both discriminative and diffusion-based baselines.
[3] The paper provides budget-aware comparisons, showing improved data efficiency.

However, please note that I am not an expert in the visual domain. Below are some of my concerns:

- Limited novelty in mechanism

The core idea of incorporating visual tokens into cross-attention is related to existing works in visual conditioning (e.g., image prompts, adapters, or token-based conditioning in diffusion models). The paper mainly adapts this idea to the UDA setting rather than introducing a fundamentally new mechanism.

- Lack of deeper analysis

The paper does not provide sufficient insight into:
[1] What information the visual tokens actually capture.
[2] Why and how they improve domain adaptation performance.
[3] Whether the tokens correspond to meaningful semantic factors.
[4] Insufficient comparison with closely related conditioning methods

It would strengthen the paper to compare more directly with existing visual-conditioning approaches (e.g., IP-Adapter-style methods) adapted to UDA.

**Requested Changes:**

How sensitive is the method to the architecture of the image-to-token encoder? Would other encoders (e.g., ViT-based) significantly affect performance?

Can the authors provide qualitative or quantitative analysis of what individual visual tokens represent?

How does the method compare to directly using image-conditioned diffusion models (e.g., ControlNet-like approaches) for the same UDA pipeline?

Is there any evidence that the method reduces domain gap at the feature level, or is the improvement purely due to better synthetic data diversity?

---

> ### Author Response · Authors · 2026-04-30
> **Responses to Reviewer GmZJ**
>
> We thank Reviewer GmZJ for the helpful feedback from the domain-adaptation perspective. The revised manuscript positions the contribution more precisely as a problem-specific conditioning framework for diffusion-guided UDA, and adds diagnostic evidence about conditioning baselines, encoder sensitivity, token responses, label consistency, and feature-level behavior.
>
> 1. **Mechanism novelty and positioning relative to existing visual conditioning.**
>
> Our main contribution is problem-specific: it introduces visual-token conditioning into diffusion-guided UDA, evaluates the generated data by downstream target-domain utility, and unifies the same token-augmented interface across source/target diffusion adaptation, pure-noise target-style synthesis, DDIM-inversion-based translation, and inference-time manipulation, while keeping the backbone, VAE, and denoising objective unchanged.
>
> This positioning is important because the objective differs from general image-prompting. In UDA, the central question is not whether generation is visually controllable in isolation, but whether the generated labeled target-style data improves target-domain classification. The main tables and the budget-aware inversion-free comparison support that goal, and the new diagnostics explain why the conditioning design matters beyond simply adding an image feature.
>
> 2. **What visual tokens capture and whether individual tokens are semantic factors.**
>
> The revised manuscript separates effectiveness from interpretability. It does not claim that each individual token corresponds to a stable semantic part or a human-interpretable factor. Instead, the claim is that the visual-token sequence, used jointly with text embeddings, provides instance-dependent conditioning that improves the downstream utility of generated target-style data.
>
> We added four experiments relevant to this point in Section “Additional Diagnostic Experiments” of the revised manuscript: (i) controlled global-feature baselines show that token-specific conditioning is better than one projected global token or ten repeated copies; (ii) target-side-token ablation shows that training the target branch with token-augmented conditioning is useful; (iii) same-class source-instance diversity shows that changing the source instance while holding the class prompt fixed changes generated target-style samples; and (iv) cross-attention heatmaps show that different visual tokens can induce different spatial attention responses. These analyses support sequence-level instance conditioning, while explicitly avoiding the stronger claim of token-level semantic disentanglement.
>
> 3. **Comparison with IP-Adapter-style visual-conditioning methods.**
>
> Please, refer to point 4 in the response to reviewer oiS8.
>
> To further clarify, we consider IP-Adapter-style methods as an important related family of stronger general-purpose image-conditioning modules, but not as the most controlled diagnostic comparison for the central claim of VT-DUDA. The revised discussion now separates these two points: IP-Adapter is relevant background for image-conditioned diffusion, while the added global-feature baselines directly test whether the proposed lightweight tokenization design itself matters for diffusion-guided UDA.
>
> The added single-global-token and repeated-global-token baselines are therefore more diagnostic for the mechanism studied here: they keep the same SDXL backbone, LoRA setting, sampler, generation budget, and ELS downstream training protocol, and only alter how the source image condition is represented. Since VT-DUDA outperforms both baselines on Ar→Cl and Cl→Rw, the result supports the specific value of token-specific conditioning under a controlled UDA setting.
>
> 4. **Sensitivity to the image-to-token encoder architecture.**
>
> To address this point, we performed new experiments comparing ResNet-18 versus ViT-Tiny encoder comparison under the inversion-free Office-Home setting with ELS. The results are reported and commented on in Section “Additional Diagnostic Experiments” of the revised appendix. The diffusion backbone, LoRA strategy, token count, sampler, generation budget, and downstream UDA protocol are held fixed; only the image-to-token encoder backbone changes.
>
> **Table: Sensitivity to the image-to-token encoder architecture on Office-Home under the inversion-free protocol with ELS.**
>
> | Method | Ar→Cl | Cl→Rw |
> |---|---:|---:|
> | ELS | 57.79 | 76.43 |
> | ELS+VT-DUDA (ResNet-18 encoder) | **59.24** | 83.30 |
> | ELS+VT-DUDA (ViT-Tiny encoder) | 58.71 | **83.68** |
>
> The two encoders produce similar results: ViT-Tiny is 0.53 points lower on Ar→Cl but 0.38 points higher on Cl→Rw. Both improve over ELS. This suggests that VT-DUDA is not narrowly tied to ResNet-18; the default ResNet-18 choice is retained because it is lightweight and competitive.

---

> > ### Author Response · Authors · 2026-04-30
> > **Responses to Reviewer GmZJ**
> >
> > 5. **Relation to ControlNet-like dense image-conditioned diffusion.**
> >
> > ControlNet-like approaches can be regarded as a different design point. They usually introduce an additional control branch and rely on structured spatial controls such as edges, depth, pose, or segmentation. VT-DUDA instead uses compact source-image-derived tokens and standard cross-attention, without requiring extra dense annotations or a heavy external control pathway. This choice is aligned with the UDA setting, where the goal is not exact spatial control but efficient construction of useful labeled target-style data.
> >
> > For this reason, ControlNet is not a simple drop-in replacement for the present method. Comparing a dense-control pipeline would simultaneously vary the control signal, architecture, compute, and possibly external supervision, while our aim is to prove that our designed concept-token are beneficial for improving performance on the UDA downstream task.
> >
> > 6. **Feature-level evidence for domain-gap reduction versus synthetic data diversity.**
> >
> > To clarify this point, we refer to the t-SNE visualization reported on Office-Home: generated target-style samples occupy a region closer to the unlabeled target-domain features than the original source-domain samples. This supports the interpretation that the generated data is not merely more diverse, but also more target-like in the downstream feature space.
> >
> > VT-DUDA does not explicitly optimize a feature-alignment loss inside the diffusion model; it constructs synthetic target-style supervision through stronger conditioning. Therefore target-domain classification accuracy remains the primary criterion, while t-SNE and the new diagnostics explain the mechanism from complementary angles: target-like feature placement, source-instance-dependent generation, label-consistency behavior, and improved controlled conditioning baselines.

---

### Review · Reviewer_oiS8 · 2026-04-02

**Summary Of Contributions:**

This paper proposes VT-DUDA, a visual-token conditioning framework for diffusion-guided unsupervised domain adaptation (UDA). The core idea addresses a limitation in existing diffusion-based UDA methods: class-level text prompts provide only coarse guidance, causing different source instances of the same class to share identical conditioning signals. VT-DUDA augments text conditioning with instance-level visual tokens extracted from source images via a learned image-to-token encoder (ResNet-18 backbone mapping to M tokens). These visual tokens are concatenated with text embeddings and injected into the cross-attention layers of a latent diffusion model (SDXL) equipped with domain-specific LoRA adapters.

**Audience:**

Yes

**Audience Explanation:**

This paper addresses a timely topic at the intersection of diffusion models and domain adaptation. The findings would interest:

1) UDA researchers: The generation-based view of UDA and the emphasis on "downstream usefulness of synthetic data" provide a fresh perspective compared to traditional feature alignment approaches.

2) Diffusion model practitioners: The visual-token conditioning mechanism and its integration with adapter-based fine-tuning are applicable beyond UDA.

3) Transfer learning community: The budget-aware analysis and data efficiency findings have practical implications

**Broader Impact Concerns:**

Potential concerns:

1) Synthetic data quality: Generated target-style samples are used to train classifiers that may be deployed in real applications. If the visual-token conditioning preserves source-domain biases or produces misleading class-irrelevant correlations, downstream classifiers may exhibit unexpected failure modes.

2) Dual-use potential: The visual-token conditioning mechanism could be repurposed for generating misleading synthetic data (e.g., deepfakes with specific visual attributes). While this is not the paper's focus, acknowledging this risk is appropriate.

3) Environmental cost: Training SDXL-based models and generating large synthetic datasets has non-trivial carbon footprint. Reporting energy consumption would be valuable.

**Claims And Evidence:**

Yes

**Claims Explanation:**

The experimental evidence is generally convincing. The paper evaluates on three standard UDA benchmarks with multiple baselines, reports standard deviations over three runs (Appendix A.3), and includes ablation studies on key hyperparameters. The budget-aware comparisons (Table 3) are particularly compelling—demonstrating that inversion-free VT-DUDA outperforms Terra's full configuration and matches DCDM while using 4× fewer generated images per class.

However, several claims require additional support:

1) Label assignment validity: The paper states that "this design assumes that the generated sample retains sufficient class information from the source-conditioned input" (Section 3.4). No quantitative validation of this assumption is provided. What is the classification accuracy of generated samples when evaluated by an oracle classifier? If generated samples have incorrect semantic content, the downstream classifier may learn from noisy labels.

2) Target-side visual token utility: The paper claims that target-side visual tokens during training help "align the token-conditioned interface used during training and generation" (Section 1). However, target visual tokens are paired with generic prompts ("An image") that carry no class information. An ablation removing target-side visual tokens during training would clarify whether this design choice is necessary or simply adds complexity.

3)  Instance-level guidance: While the paper argues that visual tokens provide "instance-dependent conditioning beyond text alone," it would strengthen the claims to show that different source instances of the same class actually produce meaningfully different target-style samples. Currently, qualitative results show single examples per class rather than demonstrating within-class diversity.

**Requested Changes:**

Add comparison with IP-Adapter (Ye et al., 2023): IP-Adapter is the most directly relevant prior work, also injecting image features into cross-attention for conditioning. The paper cites IP-Adapter but provides no experimental comparison. At minimum, compare:

1) IP-Adapter with source images as conditioning for target-style generation
2) Discuss architectural differences and why VT-DUDA's simpler encoder (ResNet-18 vs. CLIP) is sufficient or preferable for UDA

Also, report the classification accuracy of generated samples using a held-out classifier or the downstream classifier trained on real data. This would quantify the "label noise" in the synthetic training set and strengthen the paper's central claim about conditioning quality improving downstream utility. Also, report computational costs: Include training time, inference time per sample, and GPU memory for VT-DUDA vs. Terra and DCDM. Generation-based UDA methods have practical deployment considerations that readers need to evaluate.

---

> ### Author Response · Authors · 2026-04-30
> **Responses to Reviewer oiS8**
>
> We thank Reviewer oiS8 for the detailed feedback and practical suggestions. We added new experiments to serve as diagnostics for synthetic-label consistency, evaluation of target-side visual-token utility, within-class source-instance diversity, clarification on computational cost, and broader-impact limitations. We further clarified the relationship to IP-Adapter and better justified our experimental setup.
>
> 1. **Validity of inherited labels and post-hoc oracle label consistency.**
>
> We performed new experiments to evaluate post-hoc semantic-consistency, as suggested. The evaluator is never used for diffusion training, generation, or downstream UDA training. It is trained only after generation on real labeled images from the target domain and used solely to evaluate whether generated samples are classified as their inherited source labels.
>
> **Table: Post-hoc label-consistency analysis of generated Office-Home samples. For each transfer, an independent ResNet-50 oracle is trained on real images from the corresponding target domain and used only for evaluation.**
>
> | Generated set | Target-domain evaluator | LC (%) | ELS+VT-DUDA Acc. (%) |
> |---|---|---:|---:|
> | Ar→Cl | Clipart oracle | 54.12 | 59.24 |
> | Cl→Rw | Real-World oracle | 82.57 | 83.30 |
>
> The results obtained confirm semantic consistency for our generated images. Specifically, Cl→Rw obtains high label consistency, 82.57%, close to the corresponding downstream accuracy of 83.30%. Ar→Cl is harder and obtains a more moderate 54.12% consistency, consistent with its lower downstream accuracy for the UDA task. This confirms that inherited labels are not guaranteed to be noise-free, and the manuscript now treats label consistency as a diagnostic rather than a hidden assumption. Importantly, the evaluator uses target labels only post hoc; no target labels are used in VT-DUDA training or in the UDA training protocol.
>
> These results have been added and commented on in Section “Post-hoc Label-consistency Analysis” of the revised Appendix.
>
> 2. **Utility of target-side visual tokens during diffusion training.**
>
> To assess this point, we perform new experiments removing target-side visual tokens while keeping all other settings unchanged, in the same controlled setting as Table: Controlled conditioning ablation on Office-Home under the inversion-free protocol with ELS.
>
> The removal of target-side visual tokens reduces ELS+VT-DUDA from 59.24 to 56.82 on Ar→Cl and from 83.30 to 78.69 on Cl→Rw, with drops of 2.42 and 4.61 points, respectively.
>
> These results support the design motivation: target-side tokens are not intended to provide class supervision, since the target prompt is generic. Their role is to train the target branch under a token-augmented conditioning interface on target-domain images.
>
> The ablation shows that this interface exposure is beneficial; simply training the target branch with a generic text prompt is less effective for subsequent source-token-conditioned target-style generation.
>
> We added the results of this ablation study to Section “Additional Diagnostic Experiments” of the revised appendix.
>
> 3. **Instance-level guidance and within-class diversity.**
>
> We added new visualizations fixing the class prompt, target branch, sampler, and generation protocol, while changing only the source image used to extract visual tokens.
>
> Please, refer to Section "Cross-attention Heatmap Visualization of Visual Tokens“ of the revised appendix.
>
> The generated target-style samples exhibit visible within-class variation, which is the behavior expected from instance-dependent conditioning.
>
> This experiment complements the main quantitative results. It does not claim that each token is individually interpretable, but it directly checks the mechanism motivating VT-DUDA: different source instances of the same class should not collapse to identical class-level text conditioning. The visualization shows that the source instance affects generation under the same class prompt.

---

> > ### Author Response · Authors · 2026-04-30
> > **Responses to Reviewer oiS8**
> >
> > 4. **Relationship to IP-Adapter and why a direct IP-Adapter table is not a one-variable comparison.**
> >
> > IP-Adapter is closely related at the high level because it injects image information into diffusion generation. However, it is designed for general-purpose image prompting, whereas VT-DUDA is designed for generation-based UDA, where the evaluation target is downstream target-domain classification accuracy rather than generic image-prompt fidelity.
> >
> > A direct IP-Adapter substitution would vary several factors at once: the visual prior, the encoder pretraining, and the cross-attention architecture. IP-Adapter typically relies on a CLIP image encoder and decoupled cross-attention, whereas VT-DUDA uses a lightweight ResNet-18 image-to-token encoder and the standard token-concatenation cross-attention interface. Using CLIP would introduce a strong external vision-language prior learned from large-scale image-text data, making it difficult to attribute gains to the tokenized UDA conditioning interface itself.
> >
> > To further evaluate the impact of concept-tokens on the downstream UDA task, the revised paper incorporates several experiments as stricter controlled baselines (see Table: Controlled conditioning ablation on Office-Home under the inversion-free protocol with ELS): single-global-token and repeated-global-token conditioning use the same backbone, sampler, budget, and downstream protocol as VT-DUDA, and isolate whether token-specific conditioning matters beyond a simpler image feature.
> >
> > 5. **Synthetic-label noise and computational cost.**
> >
> > Synthetic-label noise is quantified by the label-consistency analysis in Table: Post-hoc label-consistency analysis of generated Office-Home samples, while we added a new section “Computational Cost” in the revised appendix, reporting computational cost comparing Terra UDA and VT-DUDA under our implementation on NVIDIA A100 80GB GPUs.
> >
> > We report DCDM’s larger generation budget in the experimental comparisons, but we do not report a hardware-matched runtime profile for DCDM because we do not have a directly comparable measured implementation/log under the same environment. To avoid misleading comparisons, the cost table distinguishes measured costs from budget-based comparisons.
> >
> > **Table: Diffusion-adaptation training cost. Both methods are trained for 5000 optimization steps on the same GPU class.**
> >
> > | Method | GPU | Training steps | Training time |
> > |---|---|---:|---:|
> > | Terra UDA | A100 80GB | 5000 | 98.18 h |
> > | VT-DUDA | A100 80GB | 5000 | 112.25 h |
> >
> > **Table: Runtime and GPU-memory profiling. Peak allocated and peak reserved memory are measured using CUDA memory counters. Single-image generation latency is reported in seconds per image.**
> >
> > | Metric | Terra UDA | VT-DUDA |
> > |---|---:|---:|
> > | Peak GPU memory for training | 33105.29 MiB / 32.33 GiB | 48177.98 MiB / 47.05 GiB |
> > | Peak reserved GPU memory for training | 40616.00 MiB / 39.66 GiB | 54442.00 MiB / 53.17 GiB |
> > | Pure-noise generation time per image | 1.75 s | 3.08 s |
> > | Pure-noise peak GPU memory | 9178.33 MiB / 8.96 GiB | 20372.59 MiB / 19.90 GiB |
> > | DDIM source-to-target total time per image | 3.42 s | 4.54 s |
> > | DDIM peak GPU memory | 9175.36 MiB / 8.96 GiB | 22075.05 MiB / 21.56 GiB |
> >
> > These numbers show the practical trade-off clearly. VT-DUDA improves downstream performance and sample efficiency, but it is more expensive than Terra in wall-clock training time, GPU memory, and per-image generation latency. The paper now reports this cost explicitly so that readers can evaluate the accuracy-cost trade-off rather than only the final accuracy.
> >
> > 6. **Broader-impact concerns: synthetic data quality, dual use, and environmental cost.**
> >
> > Synthetic target-style data can inherit source-domain biases or introduce class-irrelevant correlations, so any deployment-oriented use should validate the trained classifier on task-relevant real target data. The method is evaluated on academic UDA benchmarks and is not designed for identity forgery or person-generation settings, but image-conditioned generation mechanisms can be misused in broader contexts. Accordingly, we added clarifications and acknowledged this dual-use risk in the revised paper. Finally, the added computational-cost section makes the SDXL-based training and generation overhead explicit, which is a necessary step toward evaluating environmental cost.

---

> > > ### Comment · Reviewer_oiS8 · 2026-05-01
> > > **Extra experiments**
> > >
> > > My concerns are fully solved

---

### Review · Reviewer_KHbT · 2026-04-16

**Summary Of Contributions:**

**Contributions and strengths:**

1. The method addresses a real practical issue: images in the same class can still look very different, so adding visual tokens gives the model instance-specific details that a class prompt alone cannot provide.

2. The visual-token design is not only used during training. The same idea is also reused for target-style generation, translation-based augmentation, and inference-time token manipulation. One design supports several functions instead of only one narrow problem.

3. The proposed method is easy to plug in. It keeps denoiser architecture and diffusion loss, and only changes the conditioning sequence passed into cross-attention.

**Weaknesses:**

1. A core concern is that although the paper says these visual tokens give the diffusion model meaningful instance-specific guidance but do not directly prove that the tokens really preserve the important class-relevant details of each source image rather than simply acting as an extra learned conditioning signal (see below **Weaknesses in Experimental Design** for details)

2. Because the method first compresses the whole image into one global feature with a ResNet and only then converts that summary into a small set of tokens, it may lose fine-grained local details such as the exact shape of a bed frame, the wheels of a bike, and the paper does not show experiments proving those tokens still preserve fine-grained local details.

3. The paper shows changing token strength or dropping some tokens changes the generated image, which proves the tokens affect generation, but the paper also says it does not assume each token has a clear human understandable meaning (see sec. 3.5 "They do not assume that individual visual tokens are semantically disentangled or directly interpretable"), so these figures do not prove the tokens learned reliable, class-preserving visual parts or concepts from the source image. These figures only show the model is sensitive to those tokens.

4. Is the model really learning a good joint text+image conditioning, or is it mostly ignoring the text on the target side? The paper does not directly test this question.

5. The paper says training and generation use the same token format, but it's partly true. In both training and generation, the model receives text tokens and image tokens, so the format is the same. But the actual content is different. During training, the target branch sees target-image tokens together with a general prompt like “An image”. During generation, the same target branch sees source-image tokens together with a class prompt like “A bed”. So even though the input structure is the same, the information inside it is quite different.

**Audience:**

Yes

**Audience Explanation:**

Yes, it offers a simple and practical way to improve diffusion-based UDA, which is likely relevant to researchers working on domain adaptation and generative models.

**Claims And Evidence:**

Yes

**Claims Explanation:**

Generally, this paper provides solid evidence that visual tokens are useful for improving performance and function as an active conditioning signal. However, it would be beneficial to further demonstrate whether these visual tokens truly represent instance-level visual content and to clarify what information they actually capture.

**Strengths in experiments**

1. In Tables 1 and 2, VT-DUDA gets better Transfer Accuracy than both the normal UDA methods and the other diffusion-based methods on Office-Home, Office-31, and VisDA-2017. This means the proposed conditioning design seems to work well in real experiments.

2. Table 4 shows that the number of visual tokens matters. Table 4 shows evidence (using a small number of tokens works best: 5 or 10 tokens gives better results than using 20 or 50) that choosing a short, fixed token sequence is reasonable, not random.

3. Authors also provide qualitative evidence that the token-conditioned interface affects generation:

In Figure 2, they keep the text prompt the same, but make the visual tokens weaker or stronger (varying the visual token strength η). The output image changes along with the visual token strength. This proves the model is actually using visual tokens, not ignoring them.

In Figure 3, they only use some of the visual tokens instead of all of them. The translated image changes depending on which tokens are kept. This also proves the visual tokens are really affecting the generation result.

**Weaknesses in Experimental Design**

However, the above experimental results only show that the Visual-Tokens are useful for improving performance. They do not sufficiently prove that Visual-Tokens are truly meaningful visual representations. In other words, the above results show that adding Visual-Tokens helps the model, but they do not convincingly show what information the Visual-Tokens actually capture or whether the Visual-Tokens really represent instance-level visual content:

1. The ablation in Table 4 on token count only shows that using 5 or 10 tokens works better than using 20 or 50 tokens. That tells the length of the token sequence matters, but it does not tell what the tokens mean or whether they are truly representing source-instance content.

2. Figures 2 and 3 show changing token strength or selecting subsets changes the output image. This proves the tokens are being used by the generator. But the evidence that the output changes when change the tokens only proves the tokens are a conditioning signal. It does not prove the tokens store meaningful instance-level visual information.

**Evidences still required**

To prove the encoder learns meaningful instance-level visual representations, the paper still need experiment to compare VT-DUDA with a much simpler image-conditioning baseline.

For example, take one global feature vector from the source image and either use it directly as one conditioning token, or copy that same vector into multiple token slots. If VT-DUDA is clearly better, that would be stronger evidence that the tokenization design itself matters.

For example, the source image is a photo of a bed.

Baseline 1
run the same image through a backbone to get one global feature vector for the whole image (instead of producing 10 visual tokens as VT-DUDA did in Fig. 3 ) and use this single vector as one extra token concatenate it with “A bed” (instead of concatenating 10 tokens with the class prompt “A bed” as VT-DUDA did in Fig. 3 )

Baseline 2
run the same image through a backbone to get one global feature vector for the whole image (instead of producing 10 visual tokens as VT-DUDA did in Fig. 3 ) and copy or linearly project that same vector into 10 slots and concatenate them with “A bed” . The conditioning is like [text tokens; g; g; g; g; g; g; g; g; g; g], g is the same image feature repeated 10 times

**Requested Changes:**

See  Below **1. Evidences still required** and **2. Below Weaknesses**

**1. Evidences still required**

To prove the encoder learns meaningful instance-level visual representations, the paper still need experiment to compare VT-DUDA with a much simpler image-conditioning baseline.

For example, take one global feature vector from the source image and either use it directly as one conditioning token, or copy that same vector into multiple token slots. If VT-DUDA is clearly better, that would be stronger evidence that the tokenization design itself matters.

For example, the source image is a photo of a bed.

Baseline 1
run the same image through a backbone to get one global feature vector for the whole image (instead of producing 10 visual tokens as VT-DUDA did in Fig. 3 ) and use this single vector as one extra token concatenate it with “A bed” (instead of concatenating 10 tokens with the class prompt “A bed” as VT-DUDA did in Fig. 3 )

Baseline 2
run the same image through a backbone to get one global feature vector for the whole image (instead of producing 10 visual tokens as VT-DUDA did in Fig. 3 ) and copy or linearly project that same vector into 10 slots and concatenate them with “A bed” . The conditioning is like [text tokens; g; g; g; g; g; g; g; g; g; g], g is the same image feature repeated 10 times


**2. Below Weaknesses**

Weaknesses:

-A core concern is that although the paper says these visual tokens give the diffusion model meaningful instance-specific guidance but do not directly prove that the tokens really preserve the important class-relevant details of each source image rather than simply acting as an extra learned conditioning signal (see below Weaknesses in Experimental Design for details)

-Because the method first compresses the whole image into one global feature with a ResNet and only then converts that summary into a small set of tokens, it may lose fine-grained local details such as the exact shape of a bed frame, the wheels of a bike, and the paper does not show experiments proving those tokens still preserve fine-grained local details.

-The paper shows changing token strength or dropping some tokens changes the generated image, which proves the tokens affect generation, but the paper also says it does not assume each token has a clear human understandable meaning (see sec. 3.5 "They do not assume that individual visual tokens are semantically disentangled or directly interpretable"), so these figures do not prove the tokens learned reliable, class-preserving visual parts or concepts from the source image. These figures only show the model is sensitive to those tokens.

-Is the model really learning a good joint text+image conditioning, or is it mostly ignoring the text on the target side? The paper does not directly test this question.

-The paper says training and generation use the same token format, but it's partly true. In both training and generation, the model receives text tokens and image tokens, so the format is the same. But the actual content is different. During training, the target branch sees target-image tokens together with a general prompt like “An image”. During generation, the same target branch sees source-image tokens together with a class prompt like “A bed”. So even though the input structure is the same, the information inside it is quite different.

---

> ### Author Response · Authors · 2026-04-30
> **Responses to Reviewer KHbT**
>
> We thank Reviewer KHbT for the constructive and useful comments.
> To respond to the raised concerns, we provide new experiments and specifications in the next paragraphs. Specifically, we added controlled global-feature baselines, a target-side-token ablation study, experiments on label-consistency and same-class source-instance diversity, and better clarifications on token interpretability. The new experiments, results and comments have been added in the revised manuscript, and marked in blue-color for better reference.
>
>
> 1. **Controlled global-feature baselines and whether tokenization itself matters.**
>
> We performed the experiments requested by the reviewer: (i) a single-global-token baseline that projects one image-level feature as one extra conditioning token, and (ii) a repeated-global-token baseline that repeats the same global feature into ten slots. All variants use the same SDXL backbone, LoRA adaptation strategy, sampler, generation budget, and ELS downstream UDA protocol. Therefore, the comparison isolates the conditioning design as closely as possible. For the quantitative diagnostics, we use two representative Office-Home transfer tasks: Ar→Cl and Cl→Rw. Ar→Cl is a challenging transfer task, and is also used in our generation-budget ablation, while Cl→Rw is used in our token-count ablation and provides a complementary target domain. This choice allows us to evaluate the ablation results on two non-identical target styles, while remaining coherent with the rest of the ablation studies reported in the paper.
>
> **Table: Controlled conditioning ablation on Office-Home under the inversion-free protocol with ELS. All generated-data variants use the same backbone, sampler, generation budget, and downstream training protocol.**
>
> | Method | Visual conditioning | Target-side tokens | Ar→Cl | Cl→Rw |
> |---|---|---|---:|---:|
> | ELS | — | — | 57.79 | 76.43 |
> | ELS+Text-only baseline | none | no | 57.12 | 78.23 |
> | ELS+Single-global-token | one projected global feature | yes | 58.68 | 80.98 |
> | ELS+Repeated-global-token | same global feature repeated 10 times | yes | 57.26 | 79.31 |
> | ELS+VT-DUDA w/o target-side tokens | 10 token-specific visual tokens | no | 56.82 | 78.69 |
> | ELS+VT-DUDA (Ours) | 10 token-specific visual tokens | yes | **59.24** | **83.30** |
>
> Our results confirm that adding image information is useful, but not sufficient to match VT-DUDA. On Ar→Cl, VT-DUDA improves over the single-global-token baseline by 59.24-58.68=0.56 points and over the repeated-global-token baseline by 59.24-57.26=1.98 points. On Cl→Rw, the corresponding gains are 83.30-80.98=2.32 points and 83.30-79.31=3.99 points.
>
> The repeated-global-token result is especially informative: it uses the same number of visual-token positions as VT-DUDA but removes token-specific variation, with results significantly lower than VT-DUDA. Thus, the gain is not explained merely by a longer conditioning sequence or by injecting one image-level vector; but rather by token-specific instance conditioning improving the downstream utility of the generated target-style data.
>
> The new results are reported and commented on in the new section “Controlled Comparison with Simpler Image-conditioning Baselines”, added to the Appendix A, “Additional Diagnostic Experiments” of the revised paper.
>
> 2. **Fine-grained local details and the global-to-token encoder design.**
>
> We apologize for our lack of clarity on this point. VT-DUDA is not intended as a dense spatial-control method, nor as a method that reconstructs exact local parts such as bed frames or bicycle wheels. The image-to-token encoder is intentionally compact: it provides instance-level conditioning that is useful for target-style synthetic data construction, not high-fidelity local reconstruction of the source image. This distinction is important because the UDA objective is downstream target classification accuracy, not local part preservation.
>
> The empirical evidence is consistent with compact conditioning rather than dense local description. The token-count ablation in the main paper shows that M=5 or M=10 performs best, while increasing to M=20 or M=50 hurts accuracy. In the blue-marked Appendix A, “Additional Diagnostic Experiments,” the same-class source-instance diversity visualization further checks the instance-level role of the tokens: the class prompt, sampler, and target branch are fixed, and only the source image used to extract visual tokens changes. The generated target-style samples vary across rows, supporting the intended claim that the conditioning is source-instance dependent. We therefore interpret the method as learning compact instance-level guidance, not as proving full preservation of all fine-grained local details.

---

> ### Author Response · Authors · 2026-04-30
> **Responses to Reviewer KHbT**
>
> 3. **Token-strength and token-dropping figures as sensitivity evidence, not proof of human-interpretable token semantics.**
>
> We agree with the reviewer, and would like to better clarify. Figures showing token-strength scaling and token-subset selection in the originally submitted paper, aimed to show that visual-token sequence is an active conditioning signal and enables inference-time manipulation. They were not intended as evidence that each individual token corresponds to a stable human-nameable semantic part.
>
> To this end, we added cross-attention heatmap diagnostic for the ten visual tokens in the new session “Additional Diagnostic Experiments” of the revised appendix.
>
> The heatmaps show spatially non-uniform and token-dependent attention responses, suggesting that the visual-token sequence is not ignored by cross-attention. These heatmaps are informative for token-response diagnostics but should not be interpreted as proof of semantic disentanglement. Our claim is therefore deliberately limited to sequence-level hybrid conditioning improving synthetic-data utility for UDA.
>
> 4. **Whether the model learns joint text+image conditioning or ignores text on the target side.**
>
> The intended roles are complementary: the class prompt anchors the category during source-conditioned target generation, while visual tokens provide instance-dependent variation within that category.
>
> This interpretation is supported by our new experiments.
>
> First, the text-only diffusion baseline (see Table attached to previous comment) is not sufficient to match VT-DUDA.
>
> Second, post-hoc label-consistency analysis with target-domain oracle classifiers (please, refer to the first answer to reviewer oiS8) shows that many generated samples are still recognized as their inherited source labels.
>
> Third, the same-class source-instance diversity visualization (added in Section "Same-class Source-instance Diversity" of the revised appendix) keeps the class prompt fixed and changes only the source instance, showing within-class variation rather than arbitrary class replacement.
>
> Rather than proving that text and image tokens are used in a perfectly disentangled way, these experiments support a task-relevant conclusion, which is indeed the main claim of our paper: the hybrid [text tokens; visual tokens] context improves the downstream usefulness of synthetic target-style data compared with text-only or coarser image-conditioning baselines.
>
> 5. **Training-generation consistency: same interface, different information content.**
>
> We agree with the reviewer, and clarified better in the revised manuscript.
>
> During target-branch training and generation, the branch receives a mixed [text tokens; visual tokens] sequence through the same cross-attention pathway. This avoids a stronger mismatch in which the target branch is trained with text-only conditioning but receives visual tokens only at generation time.
>
> At the same time, the content is indeed phase-specific. During training, the target branch sees target-image tokens with the generic prompt “An image”; during generation, it sees source-image tokens with a class prompt such as “A bed.”
>
> The paper now states this explicitly. The claim is therefore not semantic equivalence between training and generation inputs, but consistency of the token-augmented cross-attention interface under which the target branch is trained and later used.

---

### Decision · Action_Editor_bBFv · 2026-06-02

**Recommendation:** Accept with minor revision

**Audience:**

Yes

**Audience Explanation:**

As recognized by all reviewers, the paper addresses an important problem at the intersection of diffusion models and unsupervised domain adaptation. Given the growing interest in both areas, the work is likely to attract attention from researchers and practitioners working on generative models and domain adaptation.

**Claims And Evidence:**

Yes

**Claims Explanation:**

The paper proposes a diffusion-based unsupervised domain adaptation (UDA) method, VT-DUDA, which uses visual tokens extracted from source images to provide stronger instance-level conditioning for target-style image synthesis. By injecting these visual tokens together with text embeddings into the diffusion model’s cross-attention layers, the method aims to generate more informative synthetic target-domain data. Experimental results on standard UDA benchmarks demonstrate improved target-domain accuracy over existing baselines.

The paper provides generally convincing empirical evidence for its claims through comprehensive evaluations on multiple standard UDA benchmarks. The experimental study includes comparisons with both discriminative and diffusion-based baselines, as well as ablation analyses investigating key design choices. The results consistently show that visual-token conditioning improves target-domain performance and has a measurable impact on the image generation process. In addition, the budget-aware experiments highlight the efficiency of the proposed approach.

Several reviewers noted that further analysis would strengthen the paper. In particular, additional evidence would be valuable to better understand whether the visual tokens truly capture meaningful instance-level visual semantics and what information they encode. Reviewers also suggested deeper analyses of the learned representations, more direct comparisons with simpler image-conditioning baselines and existing visual-conditioning approaches, as well as further validation of key assumptions underlying the method. The authors provided a thorough and convincing rebuttal that addressed several of these concerns and supplied additional evidence supporting the effectiveness of the proposed approach.